# Evaluating mesenchymal stem cell therapy for sepsis with preclinical meta-analyses prior to initiating a first-in-human trial

Manoj M Lalu[1,2,3], Katrina J Sullivan[2], Shirley HJ Mei[3], David Moher[2,4], Alexander Straus[2], Dean A Fergusson[2], Duncan J Stewart[3,5], Mazen Jazi[2,6], Malcolm MacLeod[7], Brent Winston[8], John Marshall[9], Brian Hutton[2], Keith R Walley[10], Lauralyn McIntyre[2,11]*, on behalf of the Canadian Critical Care Translational Biology Group

[1]Department of Anesthesiology and Pain Medicine, The Ottawa Hospital, Ottawa, Canada; [2]Clinical Epidemiology Program, The Ottawa Hospital Research Institute, Ottawa, Canada; [3]Regenerative Medicine Program, The Ottawa Hospital Research Institute, Ottawa, Canada; [4]School of Epidemiology, Public Health and Preventive Medicine, University of Ottawa, Ottawa, Canada; [5]Department of Cell and Molecular Medicine, University of Ottawa, Ottawa, Canada; [6]Faculty of Medicine, University of Ottawa, Ottawa, Canada; [7]Centre for Clinical Brain Sciences, The University of Edinburgh, Edinburgh, United Kingdom; [8]Department of Critical Care Medicine, University of Calgary, Calgary, Canada; [9]Departments of Surgery and Critical Care Medicine, Keenan Research Centre of the Li KaShing Knowledge Institute, St. Michaels Hospital, The University of Toronto, Toronto, Canada; [10]Department of Medicine, Centre for Heart Lung Innovation, University of British Columbia, Vancouver, Canada; [11]Department of Medicine, University of Ottawa, Ottawa, Canada

*For correspondence: lmcintyre@ohri.ca

**Competing interests:** The authors declare that no competing interests exist.

**Abstract** Evaluation of preclinical evidence prior to initiating early-phase clinical studies has typically been performed by selecting individual studies in a non-systematic process that may introduce bias. Thus, in preparation for a first-in-human trial of mesenchymal stromal cells (MSCs) for septic shock, we applied systematic review methodology to evaluate all published preclinical evidence. We identified 20 controlled comparison experiments (980 animals from 18 publications) of *in vivo* sepsis models. Meta-analysis demonstrated that MSC treatment of preclinical sepsis significantly reduced mortality over a range of experimental conditions (odds ratio 0.27, 95% confidence interval 0.18–0.40, latest timepoint reported for each study). Risk of bias was unclear as few studies described elements such as randomization and no studies included an appropriately calculated sample size. Moreover, the presence of publication bias resulted in a ~30% overestimate of effect and threats to validity limit the strength of our conclusions. This novel prospective application of systematic review methodology serves as a template to evaluate preclinical evidence prior to initiating first-in-human clinical studies.

**eLife digest** Most attempts to transform exciting findings from laboratories into clinical treatments are unsuccessful. One reason for this may be the failure to consider all of the laboratory work that has been performed before deciding to test a treatment on patients for the first time. In particular, negative findings (that suggest that a potential new treatment is ineffective) may be overlooked.

Stem cells may help to treat life-threatening infections, but this has not been tested in human patients. However, the effectiveness of stem cell treatments has been tested in animals that act as models of human infection.

Before deciding to begin a clinical trial of stem cell therapy for life-threatening infections, Lalu et al. performed an exhaustive search to find all the studies in which stem cells were used to treat animal models of infection. Combining the results of all of these studies using particular analysis techniques revealed that stem cell therapy increased the survival of these animals overall. These positive effects were seen over a range of different experimental conditions (for example, when treating the animals with different doses of stem cells, or giving the doses at different times).

Lalu et al. also identified some limitations with most of the laboratory studies that had tested stem cell therapy for infections. Many of the studies used animal models that may not be the best representations of humans with severe infection. In addition, many of the scientists did not report that they had used methods (such as randomization) that would generate the most confidence in their results. Despite these limitations, there was a lot of consistency in the reported results.

Overall, the results support the decision to proceed to a clinical trial that tests the effectiveness of stem cells for treating human infections. More generally, Lalu et al.'s analysis demonstrates a way of considering all laboratory evidence before deciding to proceed to a first clinical trial in humans. This may help researchers to identify promising therapies to further develop, and also to identify potential failures before they are tested in patients.

## Introduction

The decision to initiate an early phase clinical trial requires careful evaluation of the benefits and risks of a novel intervention. However, for first-in-human studies for which there is no prior clinical experience, the assessment of potential therapeutic efficacy must rely solely on the preclinical investigations. Although regulatory guidance exists for the conduct of preclinical evaluation of novel therapies (*U. S. Department of Health and Human services, 2013*), there is little guidance to help stakeholders summarize and assess the benefit and risks of novel therapies prior to first-in-human studies. As a result, the evidence from individual preclinical studies is often summarized and described in a non-systematic and potentially biased manner (*Food and Drug Administration, 2015*). Here, we present an approach to transparently evaluate preclinical evidence of a therapy prior to its potential clinical translation. Our exemplar is mesenchymal stem cell (MSC) therapy for sepsis.

A selective narrative summary of preclinical evidence has significant limitations because the methods used to identify studies are neither comprehensive nor transparent (*Sena et al., 2014*). This is of particular concern given that studies replicating high profile experiments fail in up to 50–90% of attempts (*Begley and Ellis, 2012*; *Scott et al., 2008*; *Steward et al., 2012*) and significant publication bias results in a skewed representation of effects (*Sena et al., 2010*). Further, fewer than 5% of high impact preclinical reports are clinically translated (*Contopoulos-Ioannidis et al., 2003*) and only 11% of clinically tested agents receive licensing (*Kola and Landis, 2004*). Thus trialists have based predictions of clinical success of novel therapies on flawed data and an inappropriately highly selected and positive preclinical evidence base (*Grankvist and Kimmelman, 2016*).

Systematic reviews and meta-analyses have become very popular because they can overcome many of these challenges by promoting the transparent evaluation of therapies. Systematic reviews are guided by a protocol with explicit methods to identify, synthesize (which may include meta-analysis), and appraise all investigations pertinent to a particular research question. Similarly, meta-analysis enables pooling of effect sizes across studies and increases statistical power by reducing standard error around the average effect size, providing a more precise estimate of an overall

treatment effect (*Sena et al., 2014*; *Cohn and Becker, 2003*). Systematic reviews and meta-analyses have long been regarded as essential tools to summarize and evaluate clinical research (*Higgins and Green, 2009*) and have become a requisite component of grant applications for clinical trials (*Canadian Institutes of Health Research, 2016*); however, the application of these tools to preclinical studies has been limited.

Preclinical systematic reviews may help predict the magnitude and direction of novel therapeutic effects in high stakes first-in-human trials. For example, preclinical systematic reviews of stroke (*Horn et al., 2001*) and heart failure (*Lee et al., 2003*) therapies demonstrated that the resulting negative clinical trials could have been predicted had available preclinical evidence been analyzed in a rigorous manner. Thus, thousands of patients may have avoided exposure to potential risk without any benefit (*Kalra et al., 2002*; *Shuaib et al., 2007*). Similarly, previous preclinical systematic reviews have demonstrated that failure to report threats to methodological quality (i.e. internal validity, risk of bias) and construct validity (i.e. extent a model corresponds to the human condition it is intended to represent [*Henderson et al., 2013*]) influence treatment effect sizes (*Crossley et al., 2008*; *Hirst et al., 2014*; *Macleod et al., 2008*, *2015*; *Rooke et al., 2011*). Unlike this 'retrospective' approach that has been described in previous studies, a prospective application of preclinical systematic review methodology may help delineate the limits of a therapy prior to first-in-human application.

Our preclinical systematic review was conducted prior to the initiation of a Phase 1/2 clinical trial of immunomodulatory cell therapy (mesenchymal stromal cells, mesenchymal stem cells [MSCs], "adult stem cells") for septic shock (NCT02421484). The specific question addressed was: In preclinical in-vivo animal models of sepsis, what is the effect of MSC administration (compared to control treatment) on death? Septic shock is the result of an overwhelming systemic infection; it is one of the most common and acutely devastating health problems in the intensive care unit with a 90-day mortality rate of approximately 20–30% despite modern therapy (*Peake et al., 2014*; *Mouncey et al., 2015*; *Stevenson et al., 2014*). It is caused by a maladaptive mismatch between host inflammatory response and pathogenic stimuli which leads to organ failure and death. MSCs are ubiquitous cells (*da Silva Meirelles et al., 2006*) that support tissue repair and are mobilized under inflammatory conditions (*Hannoush et al., 2011*; *Rochefort et al., 2006*). Exogenously administered MSCs represent an especially attractive therapeutic for sepsis because they have antibacterial and organ protective effects, in addition to their immune modulatory functions (*Walter et al., 2014*).

We quantitatively summarized the results of all preclinical studies of MSC therapy for *in vivo* animal models of sepsis to predict effect size and establish an ethical basis for exposing high-risk patients to this novel therapy. This is the first systematic evaluation of a novel biologic therapy prior to initiating a first-in-human trial. We believe our approach serves as a roadmap to transparently evaluate a preclinical therapy prior to its potential clinical translation. This study has been written in an explicatory manner so that other preclinical and translational researchers not familiar with systematic review methodology may replicate our approach. Readers wishing to replicate our approach for their research agendas are directed to the methods section where explanations are provided in greater depth, and encouraged to contact the authors for further guidance.

## Results

### Search results and study characteristics

Our systematic search of MEDLINE, Embase, BIOSIS, and Web of Science yielded 3114 records. Following deduplication and screening, 18 studies were included in the review (*Figure 1*). These studies were published over a six year period (2009 to 2015) and corresponded to 20 unique experiments and involved a total of 980 animals (*Table 1*) (*Bi et al., 2010*; *Chang et al., 2012*; *Chao et al., 2014*; *Gonzalez-Rey et al., 2009*; *Hall et al., 2013*; *Kim et al., 2014*; *Krasnodembskaya et al., 2012*; *Li et al., 2012*; *Liang et al., 2011*; *Luo et al., 2014*; *Mei et al., 2010*; *Nemeth et al., 2009*; *Pedrazza et al., 2014*; *Sepúlveda et al., 2014*; *Yang et al., 2015*; *Zhao et al., 2013*, *2014*; *Zhou et al., 2014*). Six authors were contacted for additional information and all replied.

All experiments used rodents, and most were mice (80%). Several methods were used to establish sepsis or sepsis-like pathophysiology, including cecal-ligation and puncture (50%), live bacterial injection (10%), and bacterial component injection (40%). Tissue sources of MSCs included bone marrow (60%), adipose tissue (20%), and umbilical cord (20%). Similarly, immunological compatibility

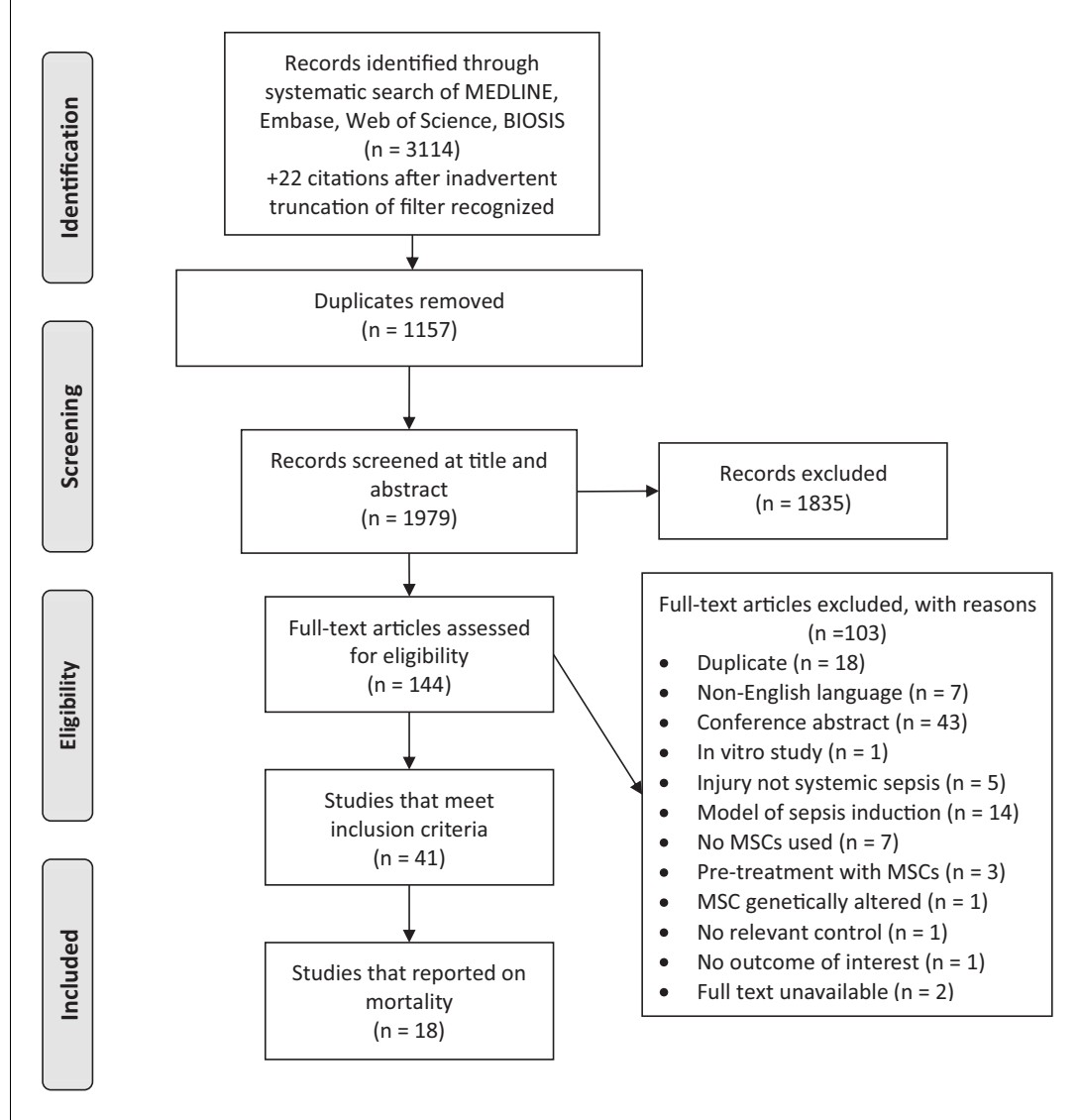

**Figure 1.** Preferred reporting items for systematic reviews and meta-analysis (PRISMA) flow diagram for study selection.

between donor MSCs and recipients varied between xenogenic (50%), syngeneic (40%), allogeneic (5%) and autologous (5%). Two of ten experiments with xenogenic cells used immunocompromised mice, while the remainder used immunocompetent mice. Total doses of MSCs ranged from 2.5 $\times$ $10^5$ to 5.0 $\times$ $10^6$ and most studies administered cells as a single dose (90%) either intravenously (80%) or intraperitoneally (20%). MSC therapy was initiated between 0 to 6 hr after experimental induction of the disease state.

## Effect of MSCs on sepsis mortality in rodents

MSC therapy in preclinical models of sepsis significantly reduced the overall odds of death (odds ratio (OR) 0.27, 95% confidence interval (CI) 0.18–0.40 (*Figure 2*). Since it is important to consider the consistency of results between studies, we calculated the $I^2$ test, which demonstrated a low degree of heterogeneity across studies ($I^2$ = 33%). The reduction in mortality was maintained regardless of when death occurred, whether considering deaths before two days after induction of sepsis (OR 0.31, 95% CI 0.21–0.46), between two and four days (OR 0.20, 95% CI 0.11–0.38), or more than four days (OR 0.18, 95% CI 0.11–0.32) (*Figure 3*).

**Table 1.** General characteristics of preclinical studies investigating the efficacy of mesenchymal stromal cells in models of sepsis.

| Author year Country | Species, Strain, Gender | Sepsis model | Resuscitation | MSC source, Compatibility | MSC Dose | Time (hours)* | MSC route | Control |
|---|---|---|---|---|---|---|---|---|
| *Gonzalez-Rey et al. (2009)*A Spain | Mouse BALB/c, NR | CLP (1 × 22 G) | None | Adipose Xenogenic or Allogeneic | $1.0 \times 10^6$ | 4 | IP | DMEM |
| *Gonzalez-Rey et al. (2009)*B Spain | Mouse BALB/c, NR | LPS (i.p.) | None | Adipose Xenogenic | $1.0 \times 10^6$ or $3.0 \times 10^5$ | 0.5 | IP | DMEM |
| *Nemeth et al. (2009)* United States | Mouse C57BL/6, M | CLP (2 × 21 G) | Fluid and antibiotics | Bone marrow Allogeneic | $1.0 \times 10^6$ | 0 or 1 | IV | PBS or Fibroblast |
| *Bi et al. (2010)* China | Mouse C57BL/6, NR | CLP (2 × 21 G) | None | Bone marrow Xenogenic | $1.0 \times 10^6$ | 1 1 | IV | PBS |
| *Mei et al. (2010)*A Canada | Mouse C57BL/ 6J, F | CLP (1 × 22 G) | Fluid | Bone marrow Syngeneic | $2.5 \times 10^5$ | 6 | IV | NS |
| *Mei et al. (2010)*B Canada | Mouse C57BL/ 6J, F | CLP (1 × 18 G) | Fluid and antibiotics | Bone marrow Syngeneic | $2.5 \times 10^5$ | 6 | IV | NS |
| *Liang et al. (2011)* China | Rat Wistar, F | LPS (i.v.) | None | Bone marrow Syngeneic | $1.0 \times 10^6$ | 2 | IV | NS |
| *Chang et al. (2012)* China | Rat SPD, M | CLP (2 × 18 G) | None | Adipose Autologous | $3 \times 1.2 \times 10^6$ | 0.5, 6 then 18 | IP | NS |
| *Krasnodembskaya et al. (2012)*, USA | Mouse C57BL/ 6J, M | *P. aeruginosa* (i.p.) | None | Bone marrow Xenogenic | $1.0 \times 10^6$ | 1 | IV | PBS Fibroblast |
| *Li et al. (2012)* China | Rat SPD, M | LPS (i.p.) | None | Umbilical cord Xenogenic | $5.0 \times 10^5$ | 1 | IV | NS or Fibroblast |
| *Hall et al. (2013)* USA | Mouse BALB/c, M | CLP (2 × 21 G) | None | Bone marrow Syngeneic | $1 \times 5.0 \times 10^5$ $+ 2 \times 2.5 \times 10^5$ | 2 then 24 then 48 | IV | PBS or Fibroblast |
| *Zhao et al. (2013)* China | Rat SPD, F | LPS (i.v.) | None | Bone marrow Syngeneic | $2.5 \times 10^6$ | 2 | IV | NS |
| *Chao et al. (2014)* Taiwan | Rat Wistar, M | CLP (1 × 18 G) | None | Bone Marrow or Umbilical Cord Xenogenic | $5.0 \times 10^6$ | 4 | IV | PBS |
| *Kim et al. (2014)* Canada | Mouse C57BL/6, M | SEB+ (i.p) | None | Bone marrow Syngeneic | $2.5 \times 10^5$ | 3 | IV | PBS |
| *Luo et al. (2014)* China | Mouse C57Bl/6, M | CLP (2 × 21 G) | Fluid | Bone marrow Syngeneic | $1.0 \times 10^6$ | 3 | IV | NS |
| *Pedrazza et al. (2014)* Brazil | Mouse C57BL/6, M | *E. coli* (i.p.) | None | Adipose Syngeneic | $1.0 \times 10^6$ | 0 | IV | PBS |
| *Sepulveda et al. (2014)* Spain | Mouse BALB/c, M | LPS (i.p.) | None | Bone Marrow Xenogenic | $1.0 \times 10^6$ | 0.5 | IP | PBS |
| *Zhao et al. (2014)* China | Mouse C57BL/6, M | CLP (NR) | None | Umbilical cord Xenogenic | $1.0 \times 10^6$ | 1 | IV | NS |
| *Zhou et al. (2014)* China | Mouse NOD SCID, M | LPS+ (i.p.) | None | Umbilical Cord Xenogenic | $2.0 \times 10^6$ | 6 | IV | No treatment |
| *Yang et al. (2015)* China | Mouse NOD SCID, M | LPS+ (i.p.) | None | Umbilical cord Xenogenic | $5.0 \times 10^5$ | 0 | IV | DMEM |

Legend: * = Time of delivery post-sepsis induction, + = Models also administered D-galactosamine, CLP = Cecal ligation and puncture, DMEM = Dulbecco's modified Eagle's medium, i.p. = Intraperitoneal, i.v. = Intravenous, LPS = Lipopolysaccharide, NR = Not reported, NOD SCID = NOD.Cg-Prkdc$^{scid}$ Il2rg$^{tm1Wjl}$/SzJ (immunodeficient), NS = Normal saline, PBS = Phosphate buffered saline, SEB = *Staphylococcal* enterotoxin B, SPD = Sprague-Dawley.

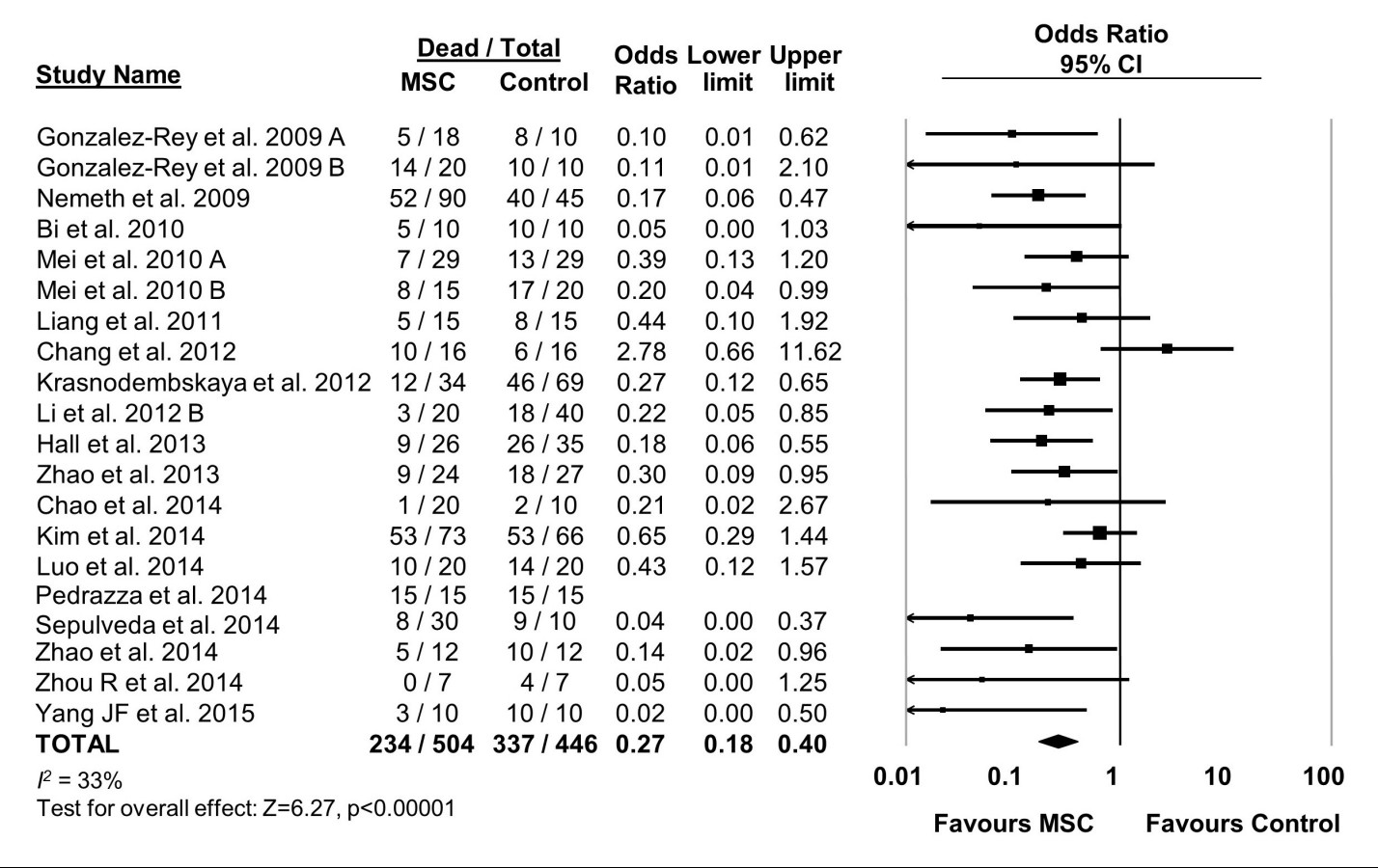

**Figure 2.** Forest plot summarizing effects of mesenchymal stromal cell (MSC) therapy on mortality in preclinical models of sepsis and endotexemia. Point estimates (odds ratio) and 95% confidence intervals (CI) are depicted for individual studies; size of point estimate depicts relative contribution to pooled effect. A pooled meta-analytic summary (random effects model) of overall effect of MSC therapy on mortality is depicted by the diamond at the bottom of the plot (vertical points represent odds ratio point estimate and horizontal points represent 95% CIs). Heterogeneity is represented with the $I^2$ statistic. Data from *Pedrazza et al. (2014)* was included in total counts but not included in meta-analysis due to 100% mortality in both study arms.

The following figure supplements are available for figure 2:

**Figure supplement 1.** Forest plot summarizing relationship of compatibility of donor mesenchymal stromal cell (MSC) with recipient animal (xenogenic vs syngeneic vs allogeneic vs autologous) on mortality in preclinical models of sepsis and endotexemia.

**Figure supplement 2.** Forest plot summarizing relationship of mesenchymal stromal cell (MSC) dose on mortality in preclinical models of sepsis and endotexemia.

**Figure supplement 3.** Forest plot summarizing relationship of mesenchymal stromal cell (MSC) therapy timing of administration on mortality in preclinical models of sepsis and endotexemia.

**Figure supplement 4.** Forest plot summarizing relationship of mesenchymal stromal cell (MSC) administration route (intravenous vs intraperitoneal) on mortality in preclinical models of sepsis and endotexemia.

**Figure supplement 5.** Forest plot summarizing relationship of mesenchymal stromal cell (MSC) tissue source (adipose vs bone marrow vs umbilical cord tissue) on mortality in preclinical models of sepsis and endotexemia.

**Figure supplement 6.** Forest plot summarizing relationship of animal species (rat vs mouse) on mortality in preclinical models of sepsis and endotexemia treated with mesenchymal stromal cells (MSCs).

**Figure supplement 7.** Forest plot summarizing relationship of animal sex (male vs female vs unreported) on mortality in preclinical models of sepsis and endotexemia treated with mesenchymal stromal cells (MSCs).

*Figure 2 continued on next page*

*Figure 2 continued*

**Figure supplement 8.** Forest plot summarizing relationship of preclinical models of sepsis and endotoxemia (cecal ligation and puncture vs live bacteria administration vs bacterial product such as lipopolysaccharide) on mortality following treatment with mesenchymal stromal cells (MSCs).

**Figure supplement 9.** Forest plot summarizing relationship of resuscitation (fluids +/- antibiotics vs no resuscitation) on mortality in preclinical models of sepsis and endotoxemia treated with mesenchymal stromal cells (MSCs).

**Figure supplement 10.** Forest plot summarizing relationship of comparison (control) treatment on mortality in preclinical models of sepsis and endotoxemia treated with mesenchymal stromal cells (MSCs).

**Figure supplement 11.** Forest plot summarizing relationship of adherence to elements of construct validity on mortality in preclinical models of sepsis and endotoxemia treated with mesenchymal stromal cells (MSCs).

## Assessment of threats to external validity/generalizability

The effects of therapies may not be sustained under varied experimental conditions, so we evaluated the generalizability and replicability of results by analyzing efficacy in pre-specified sub-groups. Heterogeneity (i.e. $I^2$ statistic) was low to moderate unless otherwise stated. Similar efficacy was noted regardless of the compatibility of donor MSCs with recipient animal (syngeneic vs. allogeneic vs. xenogenic, *Figure 2—figure supplement 1*), dose of MSC (<1.0 × 10$^6$ cells vs. ≥1.0 × 10$^6$ cells, *Figure 2—figure supplement 2*), and timing of a single dose of MSCs (less than or equal to 1 hr vs. 1–6 hr after disease induction, *Figure 2—figure supplement 3*). Intravenous administration of MSCs demonstrated efficacy (OR 0.28, 95% CI 0.20–0.40); whereas intraperitoneal administration of MSCs did not have a statistically significant effect (OR 0.21, 95% CI 0.02–1.89; *Figure 2—figure supplement 4*) and had high heterogeneity ($I^2$ = 78%), suggesting a high degree of inter-study variability. Significant effects were seen using MSCs derived from bone marrow (OR 0.13, 95% CI 0.05–0.35) and umbilical cord (OR 0.30, 95% CI 0.21–0.43; *Figure 2—figure supplement 5*), but the MSCs derived from adipose tissue did not demonstrate statistically significant efficacy (OR 0.35, 95% CI 0.03–4.39, $I^2$ = 79%). Two studies administered multiple doses of MSCs, with one demonstrating benefit and the other having no statistically significant effect. The multiple dose study with no effect was also the only investigation of autologous cells (*Chang et al., 2012*).

MSCs administered to mice were effective (OR 0.23, 95% CI 0.15–0.36) however MSC administration to rats did not produce a statistically significant effect (OR 0.47, 95% CI 0.18–1.21; *Figure 2—figure supplement 6*). Neither the sex of the diseased animal nor the model used (cecal ligation and puncture vs. live bacterial injection vs. lipopolysaccharide or other bacterial product) influenced efficacy (*Figure 2—figure supplements 7* and *8*). The addition of resuscitation (fluids or antibiotics, which are current clinical standards of therapy) did not influence the protective effect of MSCs (*Figure 2—figure supplement 9*). The comparator control group (phosphate buffered saline vs. fibroblast vs. normal saline vs. medium) had no effect; but, the one study that did not administer vehicle to the control animals did not demonstrate a statistically significant effect of MSC therapy (*Zhou et al., 2014*) (*Figure 2—figure supplement 10*).

## Assessment of threats to internal validity (methodological quality/risk of bias)

Practices such as blinding and randomization are known to affect the magnitude of effect in both clinical and preclinical studies. To determine if these threats to internal validity influenced our findings, we evaluated the risk of bias of included studies (*Table 2*). None of the experiments were considered low risk of bias across all six domains of methodological quality. Forty-five percent of experiments reported that the animals were randomized, none described methods of sequence generation or how allocation concealment was achieved. Similarly, no studies described blinding of personnel performing the experiments. One study did not blind assessors for the outcome of mortality, which may be of concern given that surrogate endpoints (i.e. not true death due to animal welfare concerns) were assessed (*Kim et al., 2014*); the remaining studies were assessed as 'unclear' as

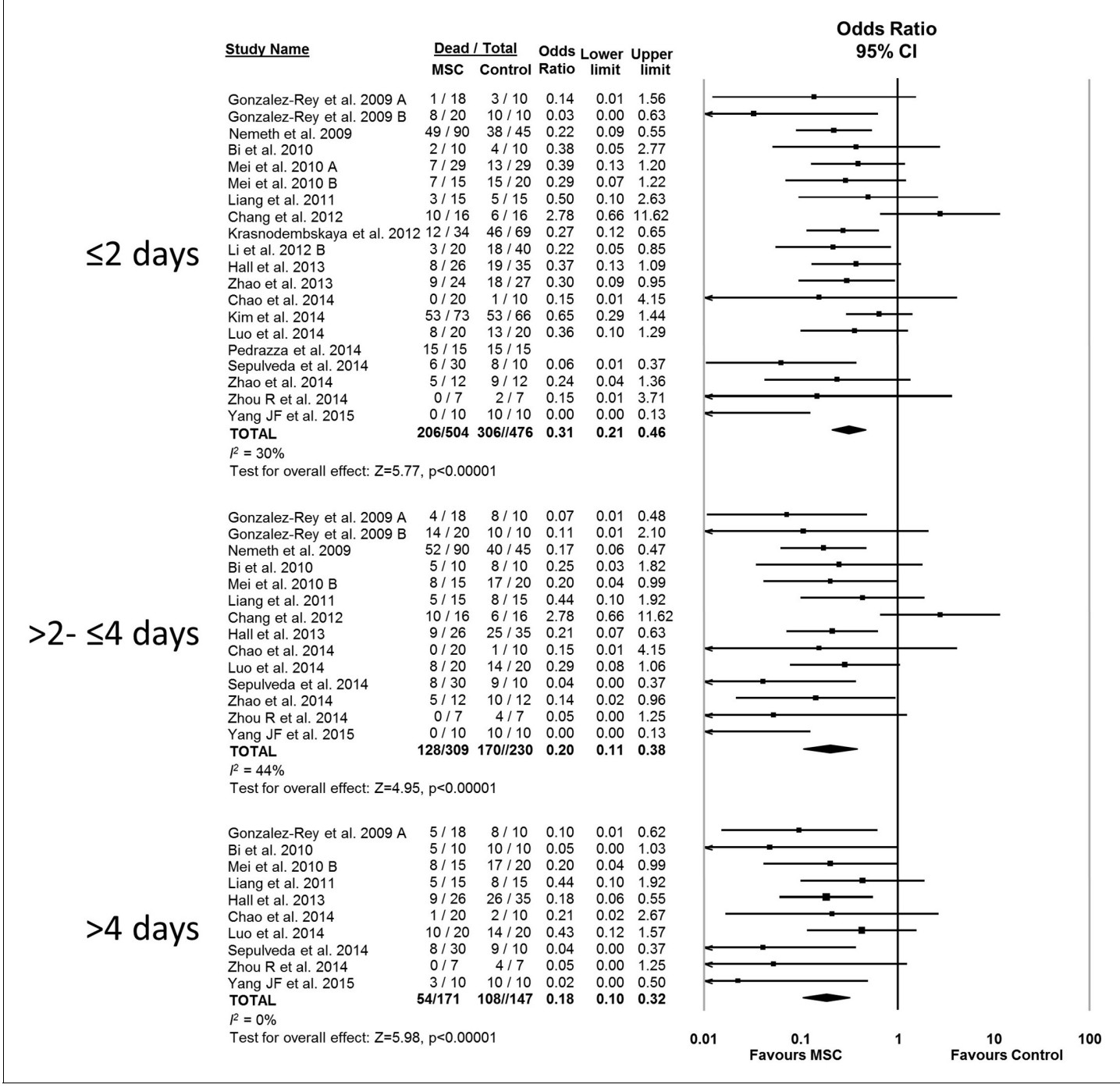

**Figure 3.** Forest plot summarizing relationship of mesenchymal stromal cell (MSC) therapy on mortality over time in preclinical models of sepsis and endotoxemia (outcome windows: ≤2 days, >2 to ≤ 4 days, > 4 days). Point estimates (odds ratio) and 95% confidence intervals (CI) are depicted for individual studies; size of point estimate depicts relative contribution to pooled effect. A pooled meta-analytic summary (random effects model) of overall effect of MSC therapy on mortality is depicted by the diamond at the bottom of each time interval (vertical points represent odds ratio point estimate and horizontal points represent 95% CIs). Heterogeneity is represented with the $I^2$ statistic. Data from *Pedrazza et al. (2014)* was included in total counts but not included in meta-analysis due to 100% mortality in both study arms.

**Table 2.** Risk of bias assessment of preclinical studies investigating the efficacy of mesenchymal stromal cells in models of sepsis.

| Study | Randomization | Allocation concealment | Blinding of personnel | Blinding of outcome assessment | Incomplete outcome data | Selective outcome reporting |
|---|---|---|---|---|---|---|
| Gonzalez-Rey et al. (2009) | U | U | U | U | L | L |
| Nemeth et al. (2009) | U | U | U | U | L | L |
| Bi et al. (2010) | U | U | U | U | H | L |
| Mei et al. (2010) | U | U | U | U | L | L |
| Liang et al. (2011) | U | U | U | U | U | L |
| Chang et al. (2012) | U | U | U | U | U | L |
| Krasnodembskaya et al. (2012) | U | U | U | U | U | L |
| Li et al. (2012) | U | U | U | U | U | L |
| Hall et al. (2013) | U | U | U | U | U | L |
| Zhao et al. (2013) | U | U | U | U | U | L |
| Chao et al. (2014) | U | U | U | U | U | L |
| Kim et al. (2014) | U | U | U | H | U | L |
| Luo et al. (2014) | U | U | U | U | U | L |
| Pedrazza et al. (2014) | U | U | U | U | U | L |
| Sepulveda 2014 | U | U | U | U | U | L |
| Zhao et al. (2014) | U | U | U | U | U | L |
| Zhou et al. (2014) | U | U | U | U | H | L |
| Yang et al. (2015) | U | U | U | U | U | L |

Legend: H = High risk of bias, L = Low risk of bias, U = Unclear risk of bias

Blinding of Outcome Assessment for Mortality: Low risk = Outcome assessors were blinded to the study groups when assessing mortality through surrogate endpoints or animals were allowed to die. Unclear = Insufficient information to determine if outcome assessors were blinded during assessment or if animals were allowed to die. High Risk = Outcome assessors not blinded to the study groups and death was defined according to surrogate endpoints.

Incomplete Outcome Data: Low risk = N values were consistent between methods and results for the mortality outcome. Unclear = N value was either not presented in the methods or in the results, and therefore there is insufficient information to permit judgement. High risk = N values were not consistent between methods and results for the mortality outcome.

Selective Reporting: Low risk = The methods section indicated mortality as a pre-specified outcome measure. High risk = The mortality outcome was presented in the results but not pre-specified in the methods section.

insufficient details of outcome assessment were reported. An assessment of high risk of bias for incomplete outcome data occurred in 10% of studies (examined as consistent *n* values reported from methods to results); in 65% of experiments the numbers (*n*) were not presented in both the methods and results in sufficient detail to permit judgment. No studies reported an appropriate rationale for selection of study sample size (where appropriate rationale included a correctly calculated sample size, *Table 3*). Given the paucity of studies that adequately implemented and reported internal validity practices, an analysis to determine the effects of high vs. low risk of bias on the effect size was not feasible.

## Assessment of threats to construct validity

It has been suggested that failed preclinical to clinical translation may be related to a mismatch between experimental conditions and the clinical disease the model is intended to represent (i.e. construct validity) (*Henderson et al., 2013*; *Kimmelman and Henderson, 2016*). To evaluate clinical generalizability of the experimental conditions used, we performed a formal evaluation of construct validity using an eight item index that had been developed in a systematic review of preclinical sepsis (*Table 4*) (*Lamontagne et al., 2010*). None of the experiments used large animal models. Two

**Table 3.** Risk of bias assessment (other domains) of preclinical studies investigating the efficacy of mesenchymal stromal cells in models of sepsis.

| Study | Baseline characteristics* | Random housing* | Source of funding | Conflict of interest | Sample size calculation |
|---|---|---|---|---|---|
| *Gonzalez-Rey et al. (2009)* | U | U | H | H | U |
| *Nemeth et al. (2009)* | U | U | L | U | U |
| *Bi et al. (2010)* | U | U | L | U | U |
| *Mei (2010)* | U | U | H | H | U |
| *Liang et al. (2011)* | U | U | L | U | U |
| *Chang et al. (2012)* | U | U | L | L | H |
| *Krasnodembskaya et al. (2012)* | U | U | L | L | U |
| *Li et al. (2012)* | U | U | L | L | U |
| *Hall et al. (2013)* | U | U | L | L | U |
| *Zhao et al. (2013)* | U | U | L | L | U |
| *Chao et al. (2014)* | U | U | L | L | U |
| *Kim et al. (2014)* | U | U | L | L | U |
| *Luo et al. (2014)* | U | U | L | L | U |
| *Pedrazza et al. (2014)* | U | U | L | L | U |
| *Sepulveda 2014* | U | U | L | L | U |
| *Zhao et al. (2014)* | U | U | L | L | U |
| *Zhou et al. (2014)* | U | U | L | L | U |
| *Yang et al. (2015)* | U | U | L | L | U |

Legend: * = Items modified from SYRCLE risk of bias tool, H = High risk of bias, L = Low risk of bias, U = Unclear risk of bias

Baseline Characteristics: Low risk = Baseline severity of disease equal between experimental groups, Unclear = Baseline severity of disease unreported, High risk = Baseline severity of disease unbalanced between experimental groups.

Random Housing: Low risk = Animal cages were randomly placed within an animal room/facility, Unclear = Housing placement unreported, High risk = Animals place in non-random arrangement in animal room/facility.

Other risk of bias was assessed according to source of funding, conflict of interest and pre-specified sample size calculations:

Source of Funding: Low risk = Non-industry source of funding (or no funding). Unclear = Funding source was not reported. High risk = Study was funded by industry.

Conflict of Interest: Low risk = Authors reported on no conflict of interest. Unclear = Conflict of interest was not reported. High risk = Authors reported on potential conflict of interests.

Sample Size Calculation: Low risk = Sample size calculations were correctly performed and followed. Unclear = Sample size calculations were not performed. High risk = Sample size calculations were incorrectly performed/followed.

experiments (10%) used animals with comorbidities (both used immunodeficient mice), 40% of experiments used adult animal models (40% did not report animal age), and 50% used infectious models of sepsis. 90% of studies initiated MSC therapy after the induction of the disease (as opposed to at the time of disease induction) but none documented severity of the disease state prior to initiating MSC therapy. Four studies used fluid resuscitation while two of these studies also administered antibiotics. Two studies incorporated a majority of construct validity elements (i.e. at least five of eight elements); there was no difference in effect size between these studies (OR 0.18, 95% CI 0.08–0.42) and those studies that incorporated fewer elements (OR 0.28, 95% CI 0.17–0.44) (*Figure 2—figure supplement 11*).

## Evidence of publication bias

For the 20 experiments, 50% demonstrated statistically significant beneficial effects of MSCs with a median sample size of 19 animals per group. Visual inspection of a funnel plot analysis of all experiments suggested that publication bias exists (*Figure 4*), which was confirmed by Egger regression (p=0.019). Post-hoc trim and fill analysis suggested a relative overestimation of effect size of 27%,

**Table 4.** Construct validity assessment of preclinical studies investigating the efficacy of mesenchymal stromal cells in models of sepsis.

| Study | Large animal model | Adult animal model | Comorbidities | Infectious model of sepsis | Therapy initiated after sepsis induction | Documented sepsis severity prior to initiating treatment | Resuscitation included antibiotics | Resuscitation included fluids |
|---|---|---|---|---|---|---|---|---|
| Gonzalez-Rey et al. (2009)A | N | N | N | Y | Y | N | N | N |
| Gonzalez-Rey et al. (2009)B | N | N | N | N | Y | N | N | N |
| Nemeth et al. (2009) | N | Y | N | Y | Y | N | Y | Y |
| Bi et al. (2010) | N | U | N | Y | Y | N | N | N |
| Mei (2010)A | N | Y | N | Y | Y | N | N | Y |
| Mei (2010)B | N | Y | N | Y | Y | N | Y | Y |
| Liang et al. (2011) | N | U | N | N | Y | N | N | N |
| Chang et al. (2012) | N | U | N | Y | Y | N | N | N |
| Krasnodembskaya et al. (2012) | N | Y | N | Y | Y | N | N | N |
| Li et al. (2012) | N | U | N | N | Y | N | N | N |
| Hall et al. (2013) | N | U | N | Y | Y | N | N | N |
| Zhao et al. (2013) | N | Y | N | N | Y | N | N | N |
| Chao et al. (2014) | N | U | N | Y | Y | N | N | N |
| Kim et al. (2014) | N | Y | N | N | Y | N | N | N |
| Luo et al. (2014) | N | U | N | Y | Y | N | N | Y |
| Pedrazza et al. (2014) | N | Y | N | Y | N | N | N | N |
| Sepulveda 2014 | N | Y | N | N | Y | N | N | N |
| Zhao et al. (2014) | N | U | N | Y | Y | N | N | N |
| Zhou et al. (2014) | N | N | Y | N | Y | N | N | N |
| Yang et al. (2015) | N | N | Y | N | N | N | N | N |

Legend: N = No, U = Unclear, Y = Yes. Letters following author and year (e.g. Mei 2010A) indicate that more than one independent experiment was conducted in the same publication.

Large Animal Model: Yes = Sheep, pig, dog, monkey. No = Mouse, rat

Adult Animal Model: Yes = Rats ≥ 6 weeks old, mice ≥ 8 weeks old. No = Rats < 6 weeks old, mice < 8 weeks old. Unclear = No age stated

Comorbidities: Yes = e.g. Diabetes, obesity, immunodeficiency. No = No comorbidities.

Infectious Model of Sepsis: Yes = Cecal-ligation and puncture, live bacterial administration. No = Bacterial product administration (e.g. lipopolysaccharide).

Therapy Initiated After Sepsis Induction: Yes = Mesenchymal stromal cells administered after sepsis model induced. No = Mesenchymal stromal cells administered at the time of sepsis induction.

Documented Sepsis Severity Prior to Initiating Treatment: Yes = Mesenchymal stromal cells administered after marker of severity (e.g. hypotension) measured. No = Mesenchymal stromal cells administered without a marker of severity being measured.

Resuscitation Included Fluids: Yes = Fluid therapy (aside from vehicle for cell administration) administered. No = Only vehicle for cell administration or no fluids administered.

although MSCs remained associated with a statistically significant reduction in mortality after adjustment (OR 0.34, 95% CI 0.22–0.52).

## Discussion

Preclinical studies provide necessary justification to conduct a first-in-human clinical trial. Thus, a systematic review approach offers an attractive method to comprehensively synthesize the totality of available evidence. Our systematic review demonstrates that MSC therapy reduces the odds of

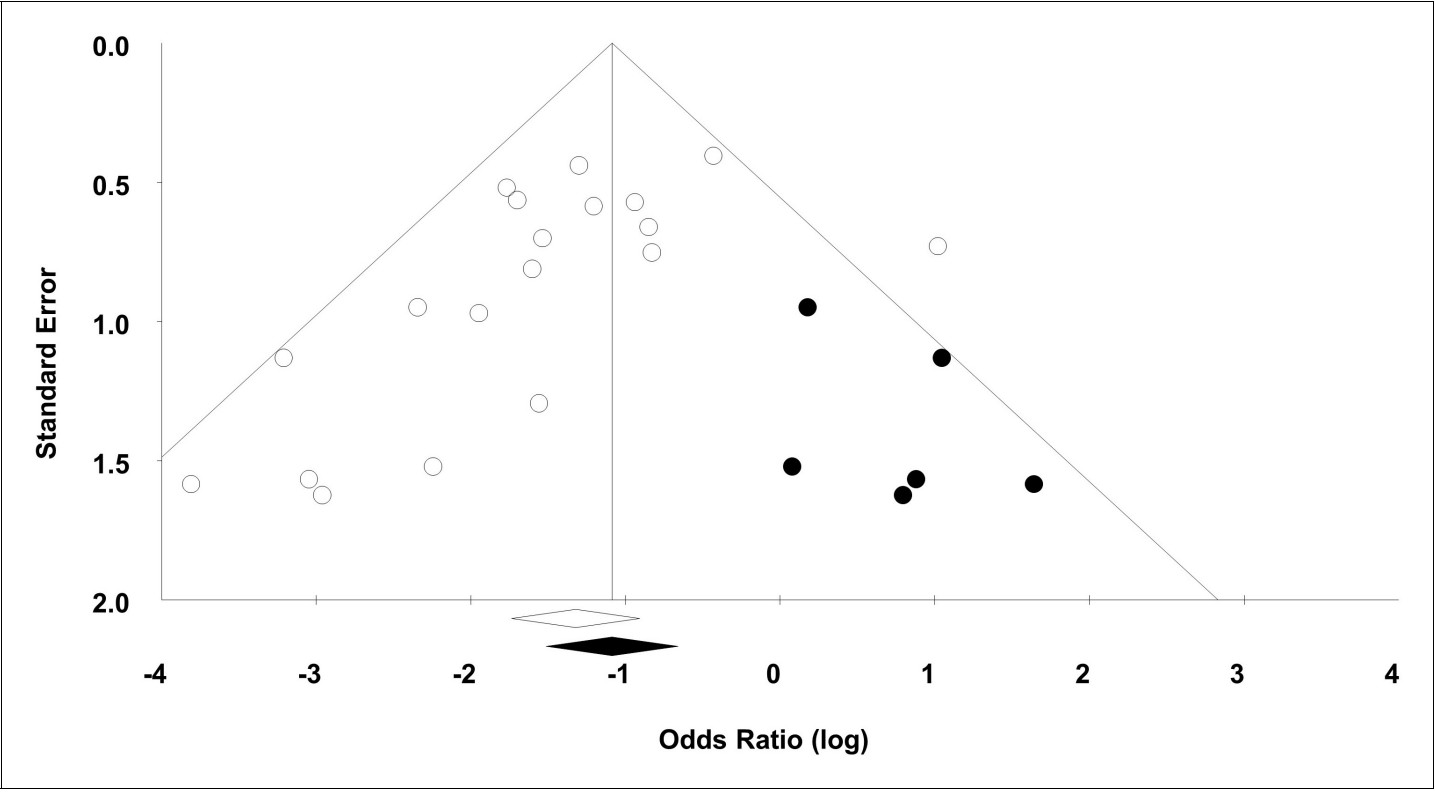

**Figure 4.** Funnel plot to detect publication bias. Trim and fill analysis was performed on overall mortality. Open circles denote original data, black circles denote 'filled' studies. Open diamond denotes original pooled effect size (log odds ratio) and 95% confidence interval. Filled diamond represents adjusted effect size and 95% confidence interval.

death in preclinical animal sepsis models. This effect is maintained over a range of time periods (less than two days, between two to four days, and longer than four days). These early outcome windows capture the majority of deaths that occur in these acute models. Moreover, the effect sizes are robustly maintained (replicated) over a variety of experimental conditions, varying models, and differing MSC immunologic compatibility (e.g. allogeneic vs. syngeneic).

It has been suggested that individual study findings have low probability of being 'true' (*Ioannidis, 2005*), however by aggregating results of similar experiments the positive predictive value of a finding dramatically increases (*Moonesinghe et al., 2007*). Thus, the findings of this systematic review helped support our decision to initiate a Phase 1/2 trial to evaluate the safety of MSC therapy in human patients with septic shock (NCT02421484). We believe our approach of systematically reviewing preclinical evidence is widely applicable for researchers considering first-in-human studies. Although our synthesis suggests MSC treatment of sepsis may be beneficial these results are tempered by the presence of potential threats to validity.

Our preclinical systematic review evaluated internal, external, and construct validity of the data. Methodological weaknesses (i.e. poor internal validity) in clinical trials are associated with an exaggeration of the treatment effect. Similarly, in preclinical studies, failure to address selection bias (through methods such as randomization and allocation concealment) and detection bias (through blinded outcome assessment) results in significantly increased effect sizes (*Crossley et al., 2008*; *Hirst et al., 2014*; *Rooke et al., 2011*). The significance of selection and detection bias has been acknowledged by The National Institutes of Health's recently issued guidelines for reporting preclinical research. These guidelines have specifically proposed randomization, blinding, and sample size calculations as key methodological information that must be described in preclinical reports (*National Institutes of Health, 2015*; *Landis et al., 2012*). In our review, none of the included studies reported randomization or allocation concealment in a manner that could be considered at low risk of bias. Similarly, no studies reported appropriate *a priori* defined sample sizes. Most of these

items were judged as 'unclear' in our risk of bias evaluation due to the convention to judge unreported items as 'unclear' rather than 'high risk'. We speculate that many of these 'unclear' items were not performed (i.e. they were 'high risk') due to a general lack of training of basic scientists in methods to reduce risk of bias (*Landis et al., 2012*; *Collins and Tabak, 2014*). This lack of reporting precluded an evaluation of their efforts and points to the need to improve the methodology used in preclinical investigations.

To address external validity (i.e. generalizability) we performed a number of subgroup analyses. Overall, subgroup analyses suggested that MSC effects appeared to be robust over a number of varying experimental conditions and across a number of different laboratories. Results of specific subgroups (e.g. autologous cells, multiple doses, intraperitoneal administration, and adipose tissue source) should be interpreted cautiously as few studies were included in these groups, and the results of one study with differing results (*Chang et al., 2012*) may have skewed data. The ability of one study to heavily influence overall effect estimates is a short-coming of meta-analyses that include few studies. As such, these subgroup analyses should be treated as exploratory.

Despite the large effect sizes noted, one must bear in mind the potential effect of publication bias (i.e. bias due to the publication of only positive studies). Our funnel plot demonstrated a highly asymmetrical pattern and our trim and fill analysis indicated that a number of unpublished negative studies may exist. This is in keeping with previous analyses of preclinical stroke data that suggested up to one in six animal studies in that field were unreported and unpublished. (*Sena et al., 2010*) Our inability to analyze these potential studies may have led to an overstatement of effect size.

To evaluate the potential clinical applicability of these results, we examined the construct validity of included studies. This was determined using recommendations that had been developed to improve the clinical generalizability of preclinical sepsis studies (*Lamontagne et al., 2010*). Animal sepsis models may not be representative of human sepsis because of the timing and severity of sepsis induction, the dose and timing of the treatment in relation to sepsis induction, the use of small/ young animals without comorbid illnesses, and lack of administration of standard of care co-interventions such as fluids and antibiotics during the study period. How well animal models of sepsis mimic the pathophysiology of human sepsis has also been a contentious issue (*Dyson and Singer, 2009*; *Osuchowski et al., 2014*; *Seok et al., 2013*). Only two studies incorporated a majority of elements addressing construct validity, thus the effect of construct validity on MSC therapy of sepsis remains to be determined.

There are a number of other issues of note that may impact the translation of MSC therapy to the clinical setting. First, although we did not formally evaluate characterization of cell products, this was variably reported in the included studies. Differences in the quality of cell therapeutics may have accounted for some of the heterogeneity of results observed. Second, dosing of cell products was not equivalent between species, even after adjusting for total cells given. Equivalence dosing of drugs between species is a complex issue and the FDA has endorsed conversion based on body surface area, rather than a dose per weight basis. (*Reagan-Shaw et al., 2008*) Applying this guidance, 1 million cells in a mouse may be equivalent to 0.5 million in a rat; similarly, this dose in a human would be roughly equivalent to 3 million cells/kg. These equivalencies should be interpreted cautiously given the differences between typical drug therapies and the cellular therapy evaluated in our review. Third, the severity of disease in these animal models at the time of MSC administration is unclear. Based on our experience with endotoxemia and cecal-ligation and puncture models, at 1 hr after disease induction some symptoms may be apparent and after 6 hr most animals have both biochemical and physiological evidence of inflammation and organ-dysfunction. Thus, we performed a subgroup analysis based on timing of administration <1 hr, >1–6 hr, >6 hr as a rough correlate to early and more delayed (intermediate and late) administration of cells in an attempt to simulate the delays in treatment that may be seen in humans who present with severe infection. Of note, no study administered cells at a late time point. A clearer reporting of disease severity at time of cell administration may allow a more precise analysis of when these cells are more (or less) efficacious. A fourth issue is the lack of transparent reporting of risk of bias elements that minimize the ability to evaluate threats to validity in our systematic review. We would suggest that general poor understanding of these core methodological issues may underlie their incomplete reporting. In order to increase the robustness and interpretation of future preclinical systematic review results we submit that authors of primary studies and journal editors should ensure adherence to published reporting guidelines for pre-clinical research studies (*National Institutes of Health , 2015*; *Kilkenny et al., 2010*). These

guidelines not only detail items relating to risk of bias (e.g. randomization and blinding) but also touch on issues that are very important when primary studies are included in systematic reviews (e.g. differentiating between biological and technical replicates, providing exact *n* numbers).

The strengths of our systematic review are in the transparent and thorough literature search and an attempt to examine potential for translation by evaluating threats to validity. To date, three clinical trials have been initiated following a systematic review and meta-analysis of animal data (*McCann et al., 2014*; *van der Worp et al., 2007*); all have repurposed currently used interventions for neurological conditions and are currently recruiting (NCT01833312, NCT01910259, ISRCTN83290762). To the best of our knowledge ours is the first preclinical systematic review that has evaluated a novel biological therapeutic in preparation for a high risk first-in-human clinical trial.

The limitations of our review should be noted. First, we restricted our search to unmodified MSCs since our group was only considering a clinical trial of unmodified cells for sepsis. Although modified cells may be of clinical interest, there are a number of additional regulatory, ethical, and safety issues which significantly increase the complexity of clinical trials using these cells; this is an issue that members of our group have experienced first-hand (NCT00936819). Other limitations of our review relate to the potential methodological limitations of the included studies. None of the included studies were considered low risk of bias across all domains, and their construct validity was highly variable. It is unclear what the influence of these methodological limitations might be in this particular study due to our inability to perform meaningful subgroup analyses. Our evaluation of the methodological aspects of included studies also relied on what the authors reported, and this may have been incomplete in cases. We would suggest however, that similar to other fields, the failure to address threats to internal validity likely contributes to an exaggerated effect size.

Despite the stated limitations of this review, the consistency of the results across the included studies and the large effect size suggest that MSCs reduce the odds of death in preclinical models of sepsis. Moreover, there are a number of studies that have demonstrated biological mechanisms that may underlie the benefits of MSCs in sepsis, including antibacterial, anti-inflammatory, and trophic effects (*Spees et al., 2016*). These mechanisms do not require engraftment and have been demonstrated to work over thousands of molecular pathways that include improved cellular energetics and activation of macrophages (*dos Santos et al., 2012*). Given the results of our review along with this biological plausibility, our group gained the support of regulatory agencies, ethics boards, and other stakeholders to proceed to a first-in-human clinical trial. Nonetheless, our efforts to translate this therapy into a clinical trial were tempered by the limitations of the preclinical studies performed to date. If this support was not provided, alternative methods to address efficacy of MSC therapy for sepsis could include conducting a low risk of bias 'confirmatory' preclinical study that was informed by the results of this systematic review (*Kleikers et al., 2015*), or performing a multicenter preclinical study (*Llovera et al., 2015*). Ultimately, ongoing and future clinical evaluations will determine whether the therapeutic effects of MSCs will translate to the human patient population.

## Materials and methods

The methods section has been written completely and transparently for researchers unfamiliar with systematic review and meta-analysis methodology. We would encourage readers wishing to replicate our approach for their own research agendas to refer to available resources (*Sena et al., 2014*; *Higgins and Green, (2009)*) and/or our group for further guidance.

### Review question and protocol

The research question for this review was, "In preclinical in-vivo animal models of sepsis, what is the effect of MSC therapy (compared to control treatment) on death?" The protocol for this review was published on the Collaborative Approach to Meta Analysis and Review of Animal Data from Experimental Studies (CAMARADES) website (http://www.dcn.ed.ac.uk/camarades/research.html#protocols) and also the University of Ottawa's Open Access Research Institutional Repository (http://hdl.handle.net/10393/32833). *A priori* publication of our protocol encourages transparency in the systematic review process and safeguards against reporting biases in the review. This review is reported in accordance with the Preferred Reporting Items for Systematic Reviews and Meta-Analysis (PRISMA) Statement (*Moher et al., 2009*). The PRISMA guidelines are an evidence-based minimum

set of items that should be reported in a systematic review and meta-analysis. Similar to other reporting guidelines, PRISMA ensures complete and transparent reporting of a study.

## Inclusion and exclusion criteria

We included all pre-clinical in vivo studies of sepsis and endotoxemia that investigated treatment with mesenchymal stromal cells. MSCs must have been administered during or after experimental induction of sepsis. Since our group was considering a clinical trial of unmodified MSCs, studies were excluded if the MSCs were differentiated, altered, or engineered to over or under express particular genes. Neonatal animal models were excluded, as were models of acute lung injury. Finally, studies where MSCs were administered with another experimental therapy or cell type were excluded.

## Literature search

To identify all relevant studies, we designed a search strategy in collaboration with a medical information specialist. We would suggest readers consult a medical librarian experienced in systematic searches if they wish to perform a literature search for a preclinical systematic review; this will ensure a comprehensive search is conducted. Although MSC terminology has been codified (*Dominici et al., 2006*) non-standard terms continue to be used in the literature, thus a number of MSC related terms were used in the search strategy. Validated animal filters were applied to increase relevancy (*de Vries et al., 2014*; *Hooijmans et al., 2010*); post-hoc, an inadvertent truncation was noted in the application of these filters, thus an updated search was performed to include the complete filters. We searched Ovid MEDLINE In-Process and Other Non-Indexed Citations, Embase Classic+Embase, BIOSIS and Web of Science (using Web of Knowledge) from inception until May 2015. The full search strategy is listed in the Appendix. Additional references were also sought through hand-searching the bibliographies of reviews and included primary studies.

## Screening

Studies were independently screened by two reviewers, with consensus required for articles to proceed to either the next screening stage or to the final analysis. Disagreements were resolved by discussion or by consultation with a senior team member when necessary.

## Data extraction

Data was extracted on the general characteristics of the study (e.g. study design, country of origin, sample size), animal model (e.g. disease induction method, use of resuscitation), and mesenchymal stromal cells (e.g. condition and source of cells). Data was collected for the primary outcome of overall mortality. Mortality was further stratified by time: $\leq$ 2 days, > 2–$\leq$ 4 days, and > 4 days. If multiple measurements were reported within a period, the latest measurement within the period was used. Data in graphical format was extracted using open source software (Engauge Digitizer, github. com; http://markummitchell.github.io/engauge-digitizer/). Extracted data were verified by a second reviewer with disagreements resolved by consultation with a third team member. Additionally, authors were contacted when further clarification was required.

## Subgroup analyses/generalizability – assessment of threats to external validity

*A priori* determined subgroup analyses were conducted to determine the effects of important factors on the estimated treatment effect. These analyses were performed to assess generalizability of results over varying experimental conditions. Subgroups were analysed for the following: animal model (e.g. mice, rat), gender, experimental model (e.g. cecal ligation and puncture, endotoxemia), source of MSC (e.g. autologous, xenogenic), route of MSC administration (e.g. intravenous, intraperitoneal), dose of MSC (less or greater than $1.0 \times 10^6$ cells), frequency of MSC dose, timing of MSC administration (less than one hour, greater than 1 hr to less than or equal to 6 hr, greater than 6 hr, or multiple dosing), resuscitation used (e.g. fluid, antibiotics), and control group (phosphate buffered saline, fibroblasts, normal saline, medium, nothing administered). Given the number of analyses performed, the results were considered exploratory and hypothesis generating. Readers employing a similar analysis may consider adjusting the value of significance based on the number of comparisons (e.g. for 11 analyses p<0.0045 would be considered significant).

### Risk of bias – assessment of threats to internal validity

Risk of bias was assessed independently in duplicate as high, low, or unclear for the six domains of bias identified by the Cochrane Risk of Bias tool (*Higgins and Green, 2009*). Domains include: (1) sequence generation, (2) allocation concealment, (3) blinding of participants and personnel, (4) blinding of outcome assessors, (5) incomplete outcome data, and (6) selective outcome reporting; operational definitions can be found in the legend for *Table 2*. Any disagreements were resolved through consultation with a senior member of the team. Other domains of risk of bias assessed were (1) source of funding, (2) conflict of interest, and (3) sample size calculations. Following reviewers' suggestions we also included the SYRCLE Risk of Bias Tool, an alternative method of assessing risk of bias in preclinical animal studies (*Hooijmans et al., 2014*). This tool is largely based on the Cochrane Risk of Bias Tool and includes several additional domains: (1) similarity of groups or adjustment for confounders at baseline, (2) random housing of animals, (3) animal selection at random for outcome assessment. The last domain was not evaluated given the outcome being assessed was death, and it was unclear for most studies whether true death or surrogate measures were being evaluated.

### Assessment of threats to construct validity

In preclinical studies construct validity refers to the extent an animal model corresponds to the clinical entity it is intended to represent (*Henderson et al., 2013*). We used a previously published framework to evaluate construct validity of the included studies (*Lamontagne et al., 2010*). Items evaluated in each study included: (1) use of a large animal model (e.g. pig, dog, sheep), (2) use of adult animals, (3) presence of co-morbid diseases, (4) use of an infectious model of sepsis, (5) documentation of severity of illness prior to initiating therapy, (6) follow-up duration $\geq$24 hr, (7) use of antibiotics, and (8) use of intravenous fluid resuscitation. Each item was assessed independently by two reviewers and assessed as either a 'yes' or a 'no'. Disagreements were resolved by consultation with a third team-member.

### Statistical analysis

Statistical analysis was performed in consultation with a statistician experienced in systematic reviews and meta-analysis. Readers seeking to replicate these methods for their own purposes are encouraged to similarly seek advice from an experienced statistician. Data from studies were pooled using meta-analysis that was performed with random effects modeling employing the DerSimonian and Laird random effects method (Comprehensive Meta-Analysis 2.0, Englewood, USA). Outcomes are expressed as odds ratios and 95% confidence intervals. There were completely independent control groups for the studies with more than one experiment extracted (i.e. a control group was not shared between two experimental groups). Thus, no correction for the number of control animals was required for multiple comparisons within a single meta-analysis. Heterogeneity of effect sizes in the overall effect estimates was assessed using the $I^2$ statistic. The following are suggested thresholds to interpret the $I^2$ statistic: 0–40% may not be important, 30–60% moderate heterogeneity, 50–90% substantial heterogeneity, 75–100% considerable heterogeneity (*Higgins and Green, 2009*).

Presence of publication bias was assessed using a funnel plot (visually) and Egger regression test (statistically). The funnel plot is a scatterplot of the intervention effect of individual studies plotted against a measure of its precision or size. The characteristic 'inverted funnel' shape arises from the fact that precision of the effect estimate increases as the as the study size increases (i.e. small studies will scatter more widely at the bottom of the funnel). A funnel plot would normally be expected to symmetrical, however the absence of symmetry can suggest publication bias (*Sterne et al., 2011*). Duval and Tweedie's trim and fill estimates were generated to estimate the number of missing studies and to estimate the adjusted effect size assuming the studies were present.

## Acknowledgements

MML was supported by a fellowship by the Heart and Stroke Foundation of Canada. DM is supported by a University Research Chair. MRM is supported by the UK NC3Rs (grant NC/L000970/1). Northern Therapeutics provided support in the form of salaries for authors [DS and SM], but did not have any additional role in the study design, data collection and analysis, decision to publish, or preparation of the manuscript. No external funding was received for this work. We wish to

acknowledge Risa Shorr, Information Specialist from the Ottawa Hospital Research Institute (OHRI) for assistance in designing the systematic search strategy and Ranjeeta Mallick statistician at the OHRI for consultation and conduct of initial statistical analysis. We also thank Dr. Tania Bubela from the University of Alberta, and Drs. AJ Frenette and Jennifer Tsang from the Canadian Critical Care Translational Biology Group for review of the manuscript.

## Additional information

### Funding

| Funder | Grant reference number | Author |
|---|---|---|
| Heart and Stroke Foundation of Canada | Fellowship | Manoj M Lalu |
| University of Ottawa | University Research Chair | David Moher |
| National Centre for the Replacement, Refinement and Reduction of Animals in Research | NC/L000970/1 | Malcolm MacLeod |

The funders had no role in study design, data collection and interpretation, or the decision to submit the work for publication.

### Author contributions

MML, Conception and design, Acquisition of data, Analysis and interpretation of data, Drafting and revision of article; KJS, Acquisition of data, Analysis and interpretation of data, Drafting and revision of article; SHJM, DM, DAF, DJS, MM, BW, JM, KRW, Conception and design, Drafting and revision of article; AS, MJ, Acquisition of data, Drafting and revision of article; BH, Analysis and interpretation of data, Drafting and revision of article; LM, Conception and design, Acquisition of data, Analysis and interpretation of data, Drafting or revising the article

### Author ORCIDs

Manoj M Lalu, http://orcid.org/0000-0002-0322-382X
Duncan J Stewart, http://orcid.org/0000-0002-9113-8691
Lauralyn McIntyre, http://orcid.org/0000-0001-7421-1407

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

# Appendix 1. Systematic search strategies

## Ovid MEDLINE(R) In-Process and Other Non-Indexed Citations and Ovid MEDLINE(R) Search Strategy:

1 exp Mesenchymal Stem Cells/

2 exp Mesenchymal Stem Cell Transplantation/

3 exp Multipotent Stem Cells/

4 exp Mesenchymal Stromal Cells/

5 ((mesenchymal adj3 (stem or stroma$one or progenitor*)) and cell$1).tw.

6 (MSC or MSCs or ADMSC or ADMSCs or BM-MSC or BM-MSCs or BMD-MSC or BMD-MSCs or BMDMSC or BMDMSCs).tw.

7 ((multipotent or multi-potent) adj3 (stroma$one cell$one or stem cell$1)).tw.

8 marrow stroma$one cell$1.tw.

9 (colony-forming unit fibroblast* or CFU-F$1).tw.

10 exp Mesoderm/cy [Cytology]

11 or/1–10

12 Stem Cell Transplantation/

13 exp Gene Therapy/

14 Mesenchymal.tw.

15 (12 or 13) and 14

16 11 or 15

17 exp Sepsis/

18 exp Bacteremia/

19 (sepsis* or septic* or pyaemi* or pyemi* or pyohemi*).tw.

20 shock.tw.

21 (fungemi* or fungaemi*or bacteremi* or bacteraemi* or endotoxemi* or endo-toxemi* or endotoxaemi* or endo-toxaemi*).tw.

22 (blood adj1 poison*).tw.

23 ((live or viable or blood or bloodstream* or clot or clots) adj3 bacter*).tw.

24 (Cecum/ or Colon,Ascending/) and ((in or su).fs. or Punctures/ or Ligation/)

25 ((Cecum or coecum or caecum or cecal or coecal or caecal) adj3 (perforat* or ligat* or punctur* or injur*)).tw.

26 (Colon adj1 ascend* adj3 (perforat* or ligat* or punctur* or injur*)).tw.

27 ((hepatic flexure or hepatic flexure) adj3 (perforat* or ligat* or punctur* or injur*)).tw.

28 ((right colic flexure or right colic flexture) adj3 (perforat* or ligat* or punctur* or injur*)).tw.

29 (proximal colon adj3 (perforat* or ligat* or punctur* or injur*)).tw.

30 colon ascendens stent peritonitis.tw.

31 (CLP or SL-CLP or CASP).tw.

32 exp systemic inflammatory response syndrome/

33 ("systemic inflammatory response" or "inflammatory response syndrome" or SIRS).tw.

34 exp lipopolysaccharides/

35 (lipopolysaccharide* or lipo-polysaccharide* or LPS or lipoglycan*).tw.

36 exp Peritonitis/

37 peritonitis.tw.

38 (exp Infection/ or exp Bacterial Infections/ or exp Inflammation/) and pp.fs.

39 exp Endotoxins/

40 (endotoxin* or ETX).tw.

41 or/17–40

42 exp animal experimentation/ or exp models, animal/ or exp animals/ or mammals/ or vertebrates/ or exp fishes/ or exp amphibia/ or exp reptiles/ or exp birds/ or exp hyraxes/ or exp marsupialia/ or exp monotremata/ or exp scandentia/ or exp chiroptera/ or exp carnivora/ or exp cetacea/ or exp Xenarthra/ or exp elephants/ or exp insectivora/ or exp lagomorpha/ or exp rodentia/ or exp sirenia/ or exp Perissodactyla/ or primates/ or exp strepsirhini/ or haplorhini/ or exp tarsii/ or exp platyrrhini/ or catarrhini/ or exp cercopithecidae/ or gorilla gorilla/ or pan paniscus/ or pan troglodytes/ or exp pongo/ or exp hylobatidae/ or hominidae/ (20563742)

43 (bombina or salientia or toad or toads or "epidalea calamita" or salamander or salamanders or eel or eels or fish or fishes or pisces or catfish or catfishes or siluriformes or arius or heteropneustes or sheatfish or perch or perches or percidae or perca or trout or trouts or char or chars or salvelinus or "fathead minnow" or minnow or cyprinidae or carps or carp or zebrafish or zebrafishes or goldfish or goldfishes or guppy or guppies or chub or chubs or tinca or barbels or barbus or pimephales or promelas or "poecilia reticulata" or mullet or mullets or seahorse or seahorses or mugil curema or atlantic cod or shark or sharks or catshark or anguilla or salmonid or salmonids or whitefish or whitefishes or salmon or salmons or sole or solea or "sea lamprey" or lamprey or lampreys or pumpkinseed or sunfish or sunfishes or tilapia or tilapias or turbot or turbots or flatfish or flatfishes or sciuridae or squirrel or squirrels or chipmunk or chipmunks or suslik or susliks or vole or voles or lemming or lemmings or muskrat or muskrats or lemmus or otter or otters or marten or martens or martes or weasel or badger or badgers or ermine or mink or minks or sable or sables or gulo or gulos or wolverine or wolverines or minks or mustela or llama or llamas or alpaca or alpacas or camelid or camelids or guanaco or guanacos or chiroptera or chiropteras or bat or bats or fox or foxes or iguana or iguanas or xenopus laevis or parakeet or parakeets or parrot or parrots or donkey or donkeys or mule or mules or zebra or zebras or shrew or shrews or bison or bisons or buffalo or buffaloes or deer or deers or bear or bears or panda or pandas or "wild hog" or "wild boar" or fitchew or fitch or beaver or beavers or jerboa or jerboas or capybara or capybaras).tw.

44 exp Drug Evaluation, Preclinical/

45 (preclinic* or pre-clinic*).tw.

46 or/42–45

47 16 and 41 and 46

***************************

## Embase Classic+Embase Search Strategy
_______________________________________

1 exp mesenchymal stem cell/

2 exp mesenchymal stem cell transplantation/

3 exp multipotent stem cell/

4 exp mesenchymal stroma cell/

5 ((mesenchymal adj3 (stem or stroma$one or progenitor*)) and cell$1).tw.

6 (MSC or MSCs or ADMSC or ADMSCs or BM-MSC or BM-MSCs or BMD-MSC or BMD-MSCs or BMDMSC or BMDMSCs).tw.

7 ((multipotent or multi-potent) adj3 (stroma$one cell$one or stem cell$1)).tw.

8 marrow stroma$one cell$1.tw.

9 (colony-forming unit fibroblast* or CFU-F$1).tw.

10 exp mesoderm/

11 (cell or cytolog*).tw.

12 10 and 11

13 or/1–9,12

14 exp stem cell transplantation/

15 exp gene therapy/

16 Mesenchymal.tw.

17 (14 or 15) and 16

18 13 or 17

19 exp sepsis/

20 (sepsis* or septic* or pyaemi* or pyemi* or pyohemi*).tw.

21 shock.tw.

22 (fungemi* or fungaemi*or bacteremi* or bacteraemi* or endotoxemi* or endo-toxemi* or endotoxaemi* or endo-toxaemi*).tw.

23 (blood adj1 poison*).tw.

24 ((live or viable or blood or bloodstream* or clot or clots) adj3 bacter*).tw.

25 (exp cecum/ or exp ascending colon/) and (exp ligation/ or exp puncture/)

26 ((Cecum or coecum or caecum or cecal or coecal or caecal) adj3 (perforat* or ligat* or punctur* or injur*)).tw.

27 (colon adj1 ascend* adj3 (perforat* or ligat* or punctur* or injur*)).tw.

28 ((hepatic flexure or hepatic flexure) adj3 (perforat* or ligat* or punctur* or injur*)).tw.

29 ((right colic flexure or right colic flexture) adj3 (perforat* or ligat* or punctur* or injur*)).tw.

30 (proximal colon adj3 (perforat* or ligat* or punctur* or injur*)).tw.

31 colon ascendens stent peritonitis.tw.

32 (CLP or SL-CLP or CASP).tw.

33 exp systemic inflammatory response syndrome/

34 ("systemic inflammatory response" or "inflammatory response syndrome" or SIRS).tw.

35 exp lipopolysaccharide/ (74366)

36 (lipopolysaccharide* or lipo-polysaccharide* or LPS or lipoglycan*).tw.

37 exp peritonitis/

38 peritonitis.tw.

39 (exp Infection/ or exp bacterial infection/ or exp inflammation/) and exp pathophysiology/

40 exp endotoxin/

41 (endotoxin* or ETX).tw.

42 or/19–41

43 exp animal experiment/ or exp animal model/ or exp experimental animal/ or exp transgenic animal/ or exp male animal/ or exp female animal/ or exp juvenile animal/ or animal/ or chordata/ or vertebrate/ or tetrapod/ or exp fish/ or amniote/ or exp amphibia/ or mammal/ or exp reptile/ or exp sauropsid/ or therian/ or exp monotremate/ or placental mammals/ or exp marsupial/ or Euarchontoglires/ or exp Afrotheria/ or exp Boreoeutheria/ or exp Laurasiatheria/ or exp Xenarthra/ or primate/ or exp Dermoptera/ or exp Glires/ or exp Scandentia/ or Haplorhini/ or exp prosimian/ or simian/ or exp tarsiiform/ or Catarrhini/ or exp Platyrrhini/ or ape/ or exp Cercopithecidae/ or hominid/ or exp hylobatidae/ or exp chimpanzee/ or exp gorilla/ or exp orang utan/44 (animal or animals or pisces or fish or fishes or catfish or catfishes or sheatfish or silurus or arius or heteropneustes or clarias or gariepinus or fathead minnow or fathead minnows or pimephales or promelas or cichlidae or trout or trouts or char or chars or salvelinus or salmo or oncorhynchus or guppy or guppies or millionfish or poecilia or goldfish or goldfishes or carassius or auratus or mullet or mullets or mugil or curema or shark or sharks or cod or cods or gadus or morhua or carp or carps or cyprinus or carpio or killifish or eel or eels or anguilla or zander or sander or lucioperca or stizostedion or turbot or turbots or psetta or flatfish or flatfishes or plaice or pleuronectes or platessa or tilapia or tilapias or oreochromis or sarotherodon or common sole or dover sole or solea or zebrafish or zebrafishes or danio or rerio or seabass or dicentrarchus or labrax or morone or lamprey or lampreys or petromyzon or pumpkinseed or pumpkinseeds or lepomis or gibbosus or herring or clupea or harengus or amphibia or amphibian or amphibians or anura or salientia or frog or frogs or rana or toad or toads or bufo or xenopus or laevis or bombina or epidalea or calamita or salamander or salamanders or newt or newts or triturus or reptilia or reptile or reptiles or bearded dragon or pogona or vitticeps or iguana or iguanas or lizard or lizards or anguis fragilis or turtle or turtles or snakes or snake or aves or bird or birds or quail or quails or coturnix or bobwhite or colinus or virginianus or poultry or poultries or fowl or fowls or chicken or chickens or gallus or zebra finch or taeniopygia or guttata or canary or canaries or serinus or canaria or parakeet or parakeets or grasskeet or parrot or parrots or psittacine or psittacines or shelduck or tadorna or goose or geese or branta or leucopsis or woodlark or lullula or flycatcher or ficedula or hypoleuca or dove or doves or geopelia or cuneata or duck or ducks or greylag or graylag or anser or harrier or circus pygargus or red knot or great knot or calidris or canutus or godwit or limosa or lapponica or meleagris or gallopavo or jackdaw or corvus or monedula or ruff or philomachus or pugnax or lapwing or peewit or plover or vanellus or swan or cygnus or columbianus or bewickii or gull or chroicocephalus or ridibundus or albifrons or great tit or parus or aythya or fuligula or streptopelia or risoria or spoonbill or platalea or leucorodia or blackbird or turdus or merula or blue tit or cyanistes or pigeon or pigeons or columba or pintail or anas or starling or sturnus or owl or athene noctua or pochard or ferina or cockatiel or nymphicus or hollandicus or skylark or alauda or tern or sterna or teal or crecca or oystercatcher or haematopus or ostralegus or shrew or shrews or sorex or araneus or

crocidura or russula or european mole or talpa or chiroptera or bat or bats or eptesicus or serotinus or myotis or dasycneme or daubentonii or pipistrelle or pipistrellus or cat or cats or felis or catus or feline or dog or dogs or canis or canine or canines or otter or otters or lutra or badger or badgers or meles or fitchew or fitch or foumart or foulmart or ferrets or ferret or polecat or polecats or mustela or putorius or weasel or weasels or fox or foxes or vulpes or common seal or phoca or vitulina or grey seal or halichoerus or horse or horses or equus or equine or equidae or donkey or donkeys or mule or mules or pig or pigs or swine or swines or hog or hogs or boar or boars or porcine or piglet or piglets or sus or scrofa or llama or llamas or lama or glama or deer or deers or cervus or elaphus or cow or cows or bos taurus or bos indicus or bovine or bull or bulls or cattle or bison or bisons or sheep or sheeps or ovis aries or ovine or lamb or lambs or mouflon or mouflons or goat or goats or capra or caprine or chamois or rupicapra or leporidae or lagomorpha or lagomorph or rabbit or rabbits or oryctolagus or cuniculus or laprine or hares or lepus or rodentia or rodent or rodents or murinae or mouse or mice or mus or musculus or murine or woodmouse or apodemus or rat or rats or rattus or norvegicus or guinea pig or guinea pigs or cavia or porcellus or hamster or hamsters or mesocricetus or cricetulus or cricetus or gerbil or gerbils or jird or jirds or meriones or unguiculatus or jerboa or jerboas or jaculus or chinchilla or chinchillas or beaver or beavers or castor fiber or castor canadensis or sciuridae or squirrel or squirrels or sciurus or chipmunk or chipmunks or marmot or marmots or marmota or suslik or susliks or spermophilus or cynomys or cottonrat or cottonrats or sigmodon or vole or voles or microtus or myodes or glareolus or primate or primates or prosimian or prosimians or lemur or lemurs or lemuridae or loris or bush baby or bush babies or bushbaby or bushbabies or galago or galagos or anthropoidea or anthropoids or simian or simians or monkey or monkeys or marmoset or marmosets or callithrix or cebuella or tamarin or tamarins or saguinus or leontopithecus or squirrel monkey or squirrel monkeys or saimiri or night monkey or night monkeys or owl monkey or owl monkeys or douroucoulis or aotus or spider monkey or spider monkeys or ateles or baboon or baboons or papio or rhesus monkey or macaque or macaca or mulatta or cynomolgus or fascicularis or green monkey or green monkeys or chlorocebus or vervet or vervets or pygerythrus or hominoidea or ape or apes or hylobatidae or gibbon or gibbons or siamang or siamangs or nomascus or symphalangus or hominidae or orangutan or orangutans or pongo or chimpanzee or chimpanzees or pan troglodytes or bonobo or bonobos or pan paniscus or gorilla or gorillas or troglodytes).ti,ab.

45 (preclinical* or pre-clinical*).tw.

46 or/43–45

47 18 and 42 and 46

***************************

## BIOSIS Previews
______________________________________________

#1 Topic=(Mesenchymal Stem Cells) OR Topic=(Mesenchymal Stem Cell Transplantation) OR Topic=(Multipotent Stem Cells) OR Topic=(Mesenchymal Stromal Cells)

#2 Topic=(((mesenchymal NEAR/3 (stem or stroma$ or progenitor*))))

#3 Topic=((multipotent NEAR/3 (stroma$ or stem)))

#4 Topic=((multi-potent NEAR/3 (stroma$ or stem)))

#5 Topic=((MSC or MSCs or ADMSC or ADMSCs or BM-MSC or BM-MSCs or BMD-MSC or BMD-MSCs or BMDMSC or BMDMSCs))

#6Topic=(((marrow stroma cell or marrow stromal cell or marrow stroma cells or marrow stromal cells)))

#7 Topic=((colony-forming unit fibroblast* or CFU-F or CFU-Fs))

#8 Topic=((mesoderm and cytolog*))

#9 #8 OR #7 OR #6 OR #5 OR #4 OR #3 OR #2 OR #1

#10 Topic=((Stem Cell Transplantation OR Gene Therapy)) AND Topic=(Mesenchymal)

#11 #10 OR #9

#12 Topic=(Sepsis) OR Topic=(Bacteremia) OR Topic=((sepsis* or septic* or pyaemi* or pyemi* or pyohemi*))

#13 Topic=(shock) OR Topic=((fungemi* or fungaemi*or bacteremi* or bacteraemi* or endotoxemi* or endo-toxemi* or endotoxaemi* or endo-toxaemi*)) OR Topic=((blood NEAR/1 poison*))

#14 Topic=(((live or viable or blood or bloodstream* or clot or clots) NEAR/3 bacter*))

#15 Topic=(((Cecum or coecum or caecum or cecal or coecal or caecal) NEAR/3 (perforat* or ligat* or punctur* or injur*)))

#16 Topic=(((colon NEAR/1 ascend*) NEAR/3 (perforat* or ligat* or punctur* or injur*)))

#17 Topic=("hepatic flexture" NEAR/3 (perforat* or ligat* or punctur* or injur*)) OR Topic=("hepatic flexture" NEAR/3 (perforat* or ligat* or punctur* or injur*))

#18 Topic=("right colic flexture" NEAR/3 (perforat* or ligat* or punctur* or injur*)) OR Topic=("right colic flexture" NEAR/3 (perforat* or ligat* or punctur* or injur*))

#19 Topic=("proximal colon" NEAR/3 (perforat* or ligat* or punctur* or injur*))

#20 Topic=(colon ascendens stent peritonitis)

#21 Topic=((CLP or SL-CLP or CASP))

#22 Topic=(systemic inflammatory response syndrome) OR Topic=(("systemic inflammatory response" or "inflammatory response syndrome" or SIRS))

#23 Topic=(lipopolysaccharide) OR Topic=((lipopolysaccharide* or lipo-polysaccharide* or LPS or lipoglycan*))

#24 Topic=(peritonitis)

#25 Topic=(Infection* or inflammation*) AND Topic=(pathophysiology)

#26 Topic=((endotoxin* or ETX))

#27 #26 OR #25 OR #24 OR #23 OR #22 OR #21 OR #20 OR #19 OR #18 OR #17 OR #16 OR #15 OR #14 OR #13 OR #12

#28 #27 AND #11

#29 Topic=(((animal$ or chordata or vertebrate* or fish or fishes or amphibian* or amphibium* or reptile$ or bird$ or mammal* or dog or dogs or canine$ or cat or cats or hyrax* or marsupial* or monotrem* or scandentia or bat or bats or carnivor* or cetacea or edentata* or elephant* or insect or insects or insectivore or lagomorph* or rodent* or mouse or mice or murine or murinae or muridae or rat or rats or pig or pigs or piglet$ or swine or rabbit$ or sheep$ or goat$ or horse$ or equus or cow or cows or cattle or calf or calves or bovine or sirenia or ungulate$ or primate$ or prosimian* or haplorhini* or tarsiiform* or simian*or platyrrhini or catarrhini or cercopithecidae or ape or apes or hylobatidae or hominid$ or chimpanzee* or gorilla* or orangutan* or monkey or monkeys or ape or apes)))

#30 Topic=(((preclinic* or pre-clinic*)))

#31 #30 OR #29

#32 #31 AND #28

***************************

# Web Of Science
_______________________________________________

#1 Topic=(Mesenchymal Stem Cells) OR Topic=(Mesenchymal Stem Cell Transplantation) OR Topic=(Multipotent Stem Cells) OR Topic=(Mesenchymal Stromal Cells)

#2 Topic=(((mesenchymal NEAR/3 (stem or stroma$ or progenitor*))))

#3 Topic=((multipotent NEAR/3 (stroma$ or stem)))

#4 Topic=((multi-potent NEAR/3 (stroma$ or stem)))

#5 Topic=((MSC or MSCs or ADMSC or ADMSCs or BM-MSC or BM-MSCs or BMD-MSC or BMD-MSCs or BMDMSC or BMDMSCs))

#6Topic=(((marrow stroma cell or marrow stromal cell or marrow stroma cells or marrow stromal cells)))

#7 Topic=((colony-forming unit fibroblast* or CFU-F or CFU-Fs))

#8 Topic=((mesoderm and cytolog*))

#9 #8 OR #7 OR #6 OR #5 OR #4 OR #3 OR #2 OR #1

#10 Topic=((Stem Cell Transplantation OR Gene Therapy)) AND Topic=(Mesenchymal)

#11 #10 OR #9

#12 Topic=(Sepsis) OR Topic=(Bacteremia) OR Topic=((sepsis* or septic* or pyaemi* or pyemi* or pyohemi*))

#13 Topic=(shock) OR Topic=((fungemi* or fungaemi*or bacteremi* or bacteraemi* or endotoxemi* or endo-toxemi* or endotoxaemi* or endo-toxaemi*)) OR Topic=((blood NEAR/1 poison*))

#14 Topic=(((live or viable or blood or bloodstream* or clot or clots) NEAR/3 bacter*))

#15 Topic=(((Cecum or coecum or caecum or cecal or coecal or caecal) NEAR/3 (perforat* or ligat* or punctur* or injur*)))

#16 Topic=(((colon NEAR/1 ascend*) NEAR/3 (perforat* or ligat* or punctur* or injur*)))

#17 Topic=("hepatic flexture" NEAR/3 (perforat* or ligat* or punctur* or injur*)) OR Topic=("hepatic flexture" NEAR/3 (perforat* or ligat* or punctur* or injur*))

#18 Topic=("right colic flexture" NEAR/3 (perforat* or ligat* or punctur* or injur*)) OR Topic=("right colic flexture" NEAR/3 (perforat* or ligat* or punctur* or injur*))

#19 Topic=("proximal colon" NEAR/3 (perforat* or ligat* or punctur* or injur*))

#20 Topic=(colon ascendens stent peritonitis)

#21 Topic=((CLP or SL-CLP or CASP))

#22 Topic=(systemic inflammatory response syndrome) OR Topic=(("systemic inflammatory response" or "inflammatory response syndrome" or SIRS))

#23 Topic=(lipopolysaccharide) OR Topic=((lipopolysaccharide* or lipo-polysaccharide* or LPS or lipoglycan*))

#24 Topic=(peritonitis)

#25 Topic=(Infection* or inflammation*) AND Topic=(pathophysiology)

#26 Topic=((endotoxin* or ETX))

#27 #26 OR #25 OR #24 OR #23 OR #22 OR #21 OR #20 OR #19 OR #18 OR #17 OR #16 OR #15 OR #14 OR #13 OR #12

#28 #27 AND #11

#29 Topic=(((animal$ or chordata or vertebrate* or fish or fishes or amphibian* or amphibium* or reptile$ or bird$ or mammal* or dog or dogs or canine$ or cat or cats or hyrax* or marsupial* or monotrem* or scandentia or bat or bats or carnivor* or cetacea or edentata* or elephant* or insect or insects or insectivore or lagomorph* or rodent* or mouse or mice or murine or murinae or muridae or rat or rats or pig or pigs or piglet$ or swine or rabbit$ or sheep$ or goat$ or horse$ or equus or cow or cows or cattle or calf or calves or bovine or sirenia or ungulate$ or primate$ or prosimian* or haplorhini* or tarsiiform* or simian*or platyrrhini or catarrhini or cercopithecidae or ape or apes or hylobatidae or hominid* or chimpanzee* or gorilla* or orangutan* or monkey or monkeys or ape or apes)))

#30 Topic=(((preclinic* or pre-clinic*)))

#31 #30 OR #29

#32 #31 AND #28

