## [Decision Letter]

Thank you for submitting your article "Evaluating mesenchymal stem cell therapy for sepsis with preclinical meta-analyses prior to a first-in-human trial" for consideration by *eLife*. Your article has been reviewed by three peer reviewers, one of whom, M Dawn Teare (Reviewer #1), is a member of our Board of Reviewing Editors and the evaluation has been overseen by Prabhat Jha as the Senior Editor. Another individual, Kim Wever (Reviewer #2), involved in review of your submission has also agreed to reveal their identity.

The reviewers have discussed the reviews with one another and the Reviewing Editor has drafted this decision to help you prepare a revised submission.

Summary:

This is a timely paper that demonstrates how to formally bring evidence together using a systematic review in pre-clinical studies, a field where this practice is relatively uncommon. The authors hope this manuscript will serve as a blueprint for future analyses. The authors had a compelling reason to analyze the literature, as they are in the midst of a clinical trial to test the efficacy of mesenchymal stem cell therapy in sepsis. In this manuscript Lalu et al. propose a framework for a meta-analysis approach to evaluate the pre-clinical evidence for MSC therapeutic benefit in sepsis. The authors had a compelling reason to analyze the literature, as they are in the midst of a clinical trial to test the efficacy of mesenchymal stem cell therapy in sepsis. In this manuscript Lalu et al. propose a framework for a meta-analysis approach to evaluate the pre-clinical evidence for MSC therapeutic benefit in sepsis. Using 20 experiments from 18 papers, the authors conclude that MSC treatment of preclinical sepsis significantly reduced mortality, even when controlling for factors such as where MSCs were derived from, how many injections were given, the type of injection, as well as other factors. This manuscript also addresses the potential for publishing bias that may affect the validity of the conclusion. The reviewers agreed that with some revision the manuscript could serve as a valuable template for other to follow when quantifying and synthesizing the evidence distinct study reports.

Essential revisions:

1) You conclude that the presented body of preclinical evidence has supported your decision to start a clinical trial. However, based on your quality assessment you also state that the quality of this body of evidence is at best unknown (with the possibility of studies being of low quality, and a risk of having many underpowered studies) and risks of several types of bias in the studies are either high or unclear. Furthermore, you find evidence for publication bias. All of these observations may lead to an overestimation of treatment efficacy. Furthermore, the effect seems to differ between species (and admin routes). Could you explain why, in spite of these shortcomings, you feel this evidence supports your decision to proceed to a clinical trial? Why is it appropriate to assume 'Unclear' does not mean 'High risk'? Have you considered performing a high-quality, confirmatory animal study instead? Can the authors offer some hypothetical alternative results of their systematic review that would have suggested alternative courses to a first-in-human study?

2) The concerns over irreproducibility are well established and many strategies have been proposed to improve the situation. Using formal evidence synthesis is just one of them. In Ioannidis (2005) well cited article he demonstrated that the posterior probability of a statistically significant research result being true was a function of the type 1 and type 2 error but most importantly the prior probability of a true positive. He presents an enumerated example of a meta-analysis of inconclusive studies where there is very high risk of bias and in this scenario the positive predictive value following the meta-analysis may also be less than 50%. What would the authors suggest is a fair prior probability of a true positive based on their quality assessment of the 20 experiments. In some ways this links with point 1 above. Bearing in mind the quality of the research presented AND the 95% CI for the overall effect size what is the strength of the evidence in justifying the next step of the research. Maybe the authors could draw this aspect out in their discussion?

3) The characteristics table indicates that some studies contain multiple comparisons between control and experimental groups. In such cases, which of the comparisons has/have been used in the meta-analysis? And if multiple comparisons with the same control group have been included in one meta-analysis, has the number of control animals been corrected for multiple comparisons?

4) You indicate that you have used the animal search filters for Pubmed and Embase (de Vries et al., 2014 and Hooijmans et al., 2010), however based on the reported full search strategy it seems that the filters have been truncated or altered? If the list of search terms shown in the appendix is correct, I think you cannot reference the search filters in this way since they are incomplete. Also, if this is the case, I am curious as to why you chose not to use the complete filters?

5) Could you please explain why studies using modified stem cells were excluded from this review? Aren't such stem cells clinically relevant?

6) You state in the Discussion that MSCs reduce mortality over a range of time points. However, I feel this range is very narrow (3-4 days). Is this a clinically relevant range? Would you advise to perform more long-term experiments to strengthen the evidence for a long-lasting effect?

7) I have a couple of questions regarding the analyses based on the dose of stem cells. The stem cell doses are presently given in number of cells. However, wouldn't the same exact number of cells represent a different dose when given to a rat versus a mouse? Would it be more meaningful to recalculate the doses to cells/kg body weight? I am concerned about this especially since you found no effect in rats, which in theory could be because they are receiving a 10x lower dose.

8) In addition, what was the rationale for having the cut-off for cell dosage subgroups at 1*10-6 cells? Is this cut-off somehow related to clinically relevant doses? I feel that the robustness of the findings for the subgroup analyses based on dose should be tested using sensitivity analyses, considering 1) recalculating to dose per kg bw and 2) varying the cut-off on which you base the two subgroups.

9) Similarly, could you explain the rationale behind the strata chosen for timing of administration (<1h versus 1-6h versus multiple dosing)?

10) Could you explain why you chose to use the Cochrane risk of bias tool, rather than the SYRLCE risk of bias tool which was designed especially to assess risks of bias in preclinical animal studies?

11) In the Introduction section, references Crossley et al., 2008; Hirst et al., 2014; Macleod et al., 2015 and Rooke et al., 2011 assessed reporting of measures to reduce bias, not the risk of bias itself. I suggest to reword this, or include references to articles that did assess bias (although as far as reviewer #2 is aware, the largest SR assessing bias was my own, and #2 was unable to take bias into account as a subgroup variable due to insufficient reporting).

12) The text states in several places that Mantel Haenszel ORs are used, but the MH method is used to combine several strata specific or study ORs? It is not obvious why an MH OR is needed rather than a crude OR. The meta-analysis is done with a random effects model whereas the MH would be a fixed effects analysis?

13) Pedrazza et al., 2014 – this study shows the same number of deaths in treatment and control- how is it included in the meta-analysis?

14) You are not the first to have used a pre-set list of questions to assess risk of bias and threats to validity and other systematic reviews using similar standards have also poor standards of reporting. Why do you think is it common practice for publications to lack sufficient details of sample size, randomisation and allocation? Is there poor general understanding or appreciation of why it is important to transparently report these details? This could be added to the Discussion.

15) The authors demonstrated that intravenous administration of MSC showed efficacy but intraperitoneal injection did not, and BM or umbilical cord derived- MSC had a better effect than adipose tissue derived MSC. However, only one study (Chang et al. 2012) showed data in an opposite direction in Figure 2—figure supplement 4 and Figure 2—figure supplement 5. Can the authors explain/discuss how the overall efficacy of MSC can be skewed by just a single outlier result?

16) At the end of the Introduction it is stated that the study has been written in an explicatory manner so that researchers not familiar with systematic reviews may replicate the approach. The manuscript can be further improved to meet this goal by explaining concepts and including more suggestions for further reading. For example, the authors could more fully explain a funnel plot, used in Figure 4, as this type of analysis may not be known to the audience of "preclinical and translational researchers not familiar with systematic review methodology." Similarly, what is an I2 statistic? What is the role of the PRISMA guidelines?

17) The authors have clearly shown that there are many significant limitations to analyzing pre-clinical studies due to missing or unclear data. In order to guide future pre-clinical studies, the authors should attempt to provide a "wish list" of variables/data fields they would like to be accounted for in pre-clinical research going forward. This "pre-clinical study blueprint" would be useful in helping to standardize future research efforts so their results can be more easily and accurately aggregated and compared.

18) In the assessment of threats to external validity/generalizability section, two studies were identified that gave multiple injections, one study showing benefit and the other study showing no benefit. Can this discrepancy be clarified by where the MSCs were derived from, as it states earlier that adipose tissue derived MSCs had no significant effect?

---

## [Author Response]

[…]

*Essential revisions:*

*1) You conclude that the presented body of preclinical evidence has supported your decision to start a clinical trial. However, based on your quality assessment you also state that the quality of this body of evidence is at best unknown (with the possibility of studies being of low quality, and a risk of having many underpowered studies) and risks of several types of bias in the studies are either high or unclear. Furthermore, you find evidence for publication bias. All of these observations may lead to an overestimation of treatment efficacy. Furthermore, the effect seems to differ between species (and admin routes). Could you explain why, in spite of these shortcomings, you feel this evidence supports your decision to proceed to a clinical trial? Why is it appropriate to assume 'Unclear' does not mean 'High risk'? Have you considered performing a high-quality, confirmatory animal study instead? Can the authors offer some hypothetical alternative results of their systematic review that would have suggested alternative courses to a first-in-human study?*

The decision to proceed to a clinical trial required concerted deliberation by our group, however it was supported by several lines of reasoning:

i) After correcting for the overestimate of efficacy there remained a strong signal of efficacy. In addition, the upper border of the 95% confidence interval for the corrected pooled estimate of death was 0.52 with treatment, which still represents a very large treatment effect.

ii) Our systematic review results are in sharp contrast to ‘retrospective’ preclinical systematic reviews in stroke and heart failure that demonstrated variable efficacy of agents that were ultimately demonstrated to have no effect in clinical trials.

iii) Subsequent work that included members of our group demonstrated the multitude of pathways that MSCs act through in preclinical sepsis models (dos Santos, Am J Pathology 2012). Thus, there was a strong biological rationale for MSC therapy in sepsis.

Given the pooled analysis around preclinical mortality, along with our clinical safety SR previously conducted (Lalu et al. PLoS ONE 2012), we felt that a first-in-human study (conservatively designed around safety) was the next appropriate step. We felt this was an appropriate compromise between proceeding with a clinical trial on the basis of promising but imperfect animal data, versus waiting until animal data are refined/perfected but delaying the introduction of potentially effective treatment for years.

Ultimately, our decision was supported by regulators (Health Canada), our research ethics board, funding agencies (CIHR and others), and our stakeholders. Nonetheless, we agree with the reviewers that many possible pathways exist for bench-to-bedside translation. We have now discussed a high quality (low risk of bias) animal study following an SR (e.g. Kleikers et al. Sci Reports 2015) along with multicenter preclinical studies (Llovera 2015 Sci Trans Med) in our Discussion:

“Given the results of our review along with this biological plausibility, our group gained the support of regulatory agencies, ethics boards, and other stakeholders to proceed to a first-in-human clinical trial. […] Ultimately, ongoing and future clinical evaluations will determine whether the therapeutic effects of MSCs will translate to the human patient population.”

Why is it appropriate to assume 'Unclear' does not mean 'High risk'?

Based on our backgrounds in basic/preclinical science, we would agree with the reviewers’ sentiment that “unclear” is almost certainly “high risk”. However, we have opted to take a conservative approach, based on the Cochrane handbook recommendations, and label unreported/incompletely reported items as ‘unclear’. This is an important point, however, so we have now addressed this in the Discussion:

“In our review, none of the included studies reported randomization or allocation concealment in a manner that could be considered at low risk of bias. Similarly, no studies reported appropriate a priori defined sample sizes. […] This lack of reporting precluded an evaluation of their efforts and points to the need to improve the methodology used in preclinical investigations.”

*2) The concerns over irreproducibility are well established and many strategies have been proposed to improve the situation. Using formal evidence synthesis is just one of them. In Ioannidis (2005) well cited article he demonstrated that the posterior probability of a statistically significant research result being true was a function of the type 1 and type 2 error but most importantly the prior probability of a true positive. He presents an enumerated example of a meta-analysis of inconclusive studies where there is very high risk of bias and in this scenario the positive predictive value following the meta-analysis may also be less than 50%. What would the authors suggest is a fair prior probability of a true positive based on their quality assessment of the 20 experiments. In some ways this links with point 1 above. Bearing in mind the quality of the research presented AND the 95% CI for the overall effect size what is the strength of the evidence in justifying the next step of the research. Maybe the authors could draw this aspect out in their Discussion?*

We thank the reviewer for this important comment that generated a great deal of discussion in our group. The issue of false research findings that is the topic of the Ioannidis paper from 2005 (PLoS Medicine) is a very important one and is the reason our group undertook a systematic review and not a narrative review, i.e. to summarize the ‘totality’ of the pre-clinical MSC sepsis comparative efficacy studies that examined death as an outcome measure. Based on the overall effect size and the upper border of the 95% confidence interval for the pooled estimate which still suggests a large effect on death (0.52), and the consistent effects of MSCs on death in the studies included in our review, we would argue that there is sufficient rationale and justification to move toward clinical evaluation.

We completely agree with Ioannidis’s emphasis on considering bias and probabilities in the interpretation of research results and this paper certainly generated a large number of commentaries and further thoughts that followed its publication (http://journals.plos.org/plosmedicine/article/comments?id=10.1371/journal.pmed.0020124). One issue that was subsequently raised that we feel is relevant to our systematic review (and not addressed in sufficient detail by Ioannidis) is how replication of research results from independent research groups can impact the positive predictive value. In cases where there is substantial replication such as is the case in our systematic review, the positive predictive value increases significantly (Moonesinghe R PLoS Med 2007, http://journals.plos.org/plosmedicine/article/authors?id=10.1371/journal.pmed.0040028). In addition, Ioannidis’s analysis does not consider biological rationale that, in the case of MSCs therapy for sepsis, we would argue is very strong.

Based on these helpful comments from the reviewers, we have now emphasized the value of replication of results in our Discussion:

“It has been suggested that individual study findings have low probability of being ‘true’ (Ioannidis, 2005), however by aggregating results of similar experiments the positive predictive value of a finding dramatically increases (Moonesinghe, Khoury and Janssens, 2007). Thus, the findings of this systematic review helped support our decision to initiate a Phase 1/2 trial to evaluate the safety of MSC therapy in human patients with septic shock (NCT02421484).”

*3) The characteristics table indicates that some studies contain multiple comparisons between control and experimental groups. In such cases, which of the comparisons has/have been used in the meta-analysis? And if multiple comparisons with the same control group have been included in one meta-analysis, has the number of control animals been corrected for multiple comparisons?*

Fortunately, there were completely independent control groups for the studies with more than one experiment extracted (Gonzalez-Rey 2009, Mei 2010). Thus, no correction for the number of control animals was required for multiple comparisons within a single meta-analysis. This point is now mentioned in the Methods:

“There were completely independent control groups for the studies with more than one experiment extracted (i.e. a control group was not shared between two experimental groups). Thus, no correction for the number of control animals was required for multiple comparisons within a single meta-analysis.”

*4) You indicate that you have used the animal search filters for Pubmed and Embase (de Vries et al., 2014 and Hooijmans et al., 2010), however based on the reported full search strategy it seems that the filters have been truncated or altered? If the list of search terms shown in the appendix is correct, I think you cannot reference the search filters in this way since they are incomplete. Also, if this is the case, I am curious as to why you chose not to use the complete filters?*

We have clarified this issue with our information specialist and discovered this was an inadvertent truncation. We have re-run the search with the full filter in Pubmed and Embase, which retrieved an additional 22 citations, all which were excluded at the title/abstract level. This error has been documented in Methods, our updated PRISMA diagram, and our search strategy. We thank the reviewers for pointing out this important oversight.

Methods:

“Validated animal filters were applied to increase relevancy (Moher et al., 2009; Dominici et al., 2006); post-hoc, an inadvertent truncation was noted in the application of these filters, thus an updated search was performed to include the complete filters.”

*5) Could you please explain why studies using modified stem cells were excluded from this review? Aren't such stem cells clinically relevant?*

We focused on unmodified cells as this was the most feasible intervention for our potential clinical trial. In order to use modified stem cells there are a number of extra ethical, regulatory, and safety issues that significantly increase complexity and decrease feasibility. Members of our team (DJS, SM) are aware of these issues as they are leading a modified cell therapy trial for myocardial infarction (e.g. https://clinicaltrials.gov/ct2/show/NCT00936819). In the future, as we and other groups gain clinical experience with modified cells in pre-clincial models, we hope to conduct a clinical trial to assess the safety and efficacy of these cells. A preclinical SR of modified stem cells would be conducted to support or refute progressing toward a clinical trial.

In order to further justify our inclusion and exclusion criteria we have now added the following to the Methods and Discussion sections:

Methods:

“Since our group was considering a clinical trial of unmodified MSCs, studies were excluded if the MSCs were differentiated, altered, or engineered to over or under express particular genes. Neonatal animal models were excluded, as were models of acute lung injury.”

Discussion:

“The limitations of our review should be noted. First, we restricted our search to unmodified MSCs since our group was only considering a clinical trial of unmodified cells for sepsis. Although modified cells may be of clinical interest, there are a number of additional regulatory, ethical, and safety issues which significantly increase the complexity of clinical trials that employ these cells; this is an issue that members of our group have experienced first-hand (NCT00936819).”

*6) You state in the Discussion that MSCs reduce mortality over a range of time points. However, I feel this range is very narrow (3-4 days). Is this a clinically relevant range? Would you advise to perform more long-term experiments to strengthen the evidence for a long-lasting effect?*

Accurate correlations between mouse and human age are difficult, however the ~2 year life span of a mouse should be put in the perspective of an ~80 year life span of a human. From this lens, the 3-4 days in a rodent experiment is certainly in line with the most relevant outcome of sepsis measured in clinical trials (i.e. 90 day mortality). It should be also noted that deaths in these acute models occur early in our experience (i.e. <72 hours), thus the time span evaluated captures the peak of severity for these models.

This has now been addressed in our Discussion:

“Our systematic review demonstrates that MSC therapy reduces the odds of death in preclinical animal sepsis models. This effect is maintained over a range of time periods (less than 2 days, between 2 to 4 days, and longer than 4 days). These early outcome windows capture the majority of deaths that occur in these acute models.”

*7) I have a couple of questions regarding the analyses based on the dose of stem cells. The stem cell doses are presently given in number of cells. However, wouldn't the same exact number of cells represent a different dose when given to a rat versus a mouse? Would it be more meaningful to recalculate the doses to cells/kg body weight? I am concerned about this especially since you found no effect in rats, which in theory could be because they are receiving a 10x lower dose.*

*8) In addition, what was the rationale for having the cut-off for cell dosage subgroups at 1*10-6 cells? Is this cut-off somehow related to clinically relevant doses? I feel that the robustness of the findings for the subgroup analyses based on dose should be tested using sensitivity analyses, considering 1) recalculating to dose per kg bw and 2) varying the cut-off on which you base the two subgroups.*

This is an interesting and very complex issue raised by the reviewers. Attempting to equate cell therapy doses in a per kilogram basis is likely incorrect, as the FDA and others have supported equivalence drug dosing based on body surface area (i.e. rather than weight; Raegan-Shaw FASEB 2008). As an example, based on these guidelines, 1 million cells in a mouse may be equivalent to 0.5 million cells in a rat.

Similarly, using body surface area, this dose of 1 million cells in a mouse would equate to approximately 3 million cells/kg in a human, which is in line with the highest dose used in our clinical trial.

Interestingly, the dose of ~1.0x106 cells per rodent has emerged as a widely used ‘standard’ dose for MSC experiments across a variety of disease states. We are unsure where this convention arose, however it would certainly be in line with doses used in human trials to date. Thus, largely based on content expertise, this was chosen a priori by our group as a cut-off for subgroup analyses.

We have addressed this issue now in the Discussion:

“There are a number of other issues of note that may impact the translation of MSC therapy to the clinical setting… Equivalence dosing of drugs between species is a complex issue and the FDA has endorsed conversion based on body surface area, rather than a dose per weight basis.(Seok et al., 2013) Applying this guidance, 1 million cells in a mouse may be equivalent to 0.5 million in a rat; similarly, this dose in a human would be roughly equivalent to 3 million cells/kg. These equivalencies should be interpreted cautiously given the differences between typical drug therapies and the cellular therapy evaluated in our review.”

*9) Similarly, could you explain the rationale behind the strata chosen for timing of administration (<1h versus 1-6h versus multiple dosing)?*

Our main purpose here was to differentiate those studies in which cell therapy was given prior to any overt symptoms of sepsis becoming apparent vs those studies in which cell therapy was given after symptoms were likely apparent. Based on our own experience with both endotoxemia and cecal-ligation puncture models, by 1 h some symptoms of sepsis are apparent. We acknowledge that this is a rather blunt/indirect instrument to judge disease severity at the time of therapy; however this is the best we could do in the absence of any papers reporting disease severity at the time MSCs were administered. This is now detailed in our Discussion:

“Third, the severity of disease in these animal models at the time of MSC administration is unclear. […] A clearer reporting of disease severity at time of cell administration may allow a more precise analysis of when these cells are more (or less) efficacious.”

*10) Could you explain why you chose to use the Cochrane risk of bias tool, rather than the SYRLCE risk of bias tool which was designed especially to assess risks of bias in preclinical animal studies?*

Elements such as blinding and randomization are addressed in both tools and have the clearest empirical evidence of affecting preclinical effect sizes. Nonetheless, we would be remiss to not introduce the SYRCLE tool to our intended audience. As such, we have now included a SYRCLE RoB elements in the Methods section and Table 3:

“Following reviewers’ suggestions we also included the SYRCLE Risk of Bias Tool, an alternative method of assessing risk of bias in preclinical animal studies (de Vries et al., 2014). This tool is largely based on the Cochrane tool and includes several additional domains, including 1) similarity of groups or adjustment for confounders at baseline, 2) random housing of animals, 3) animal selection at random for outcome assessment.”

*11) In the Introduction section, references Crossley et al., 2008; Hirst et al., 2014; Macleod et al., 2015 and Rooke et al., 2011 assessed reporting of measures to reduce bias, not the risk of bias itself. I suggest to reword this, or include references to articles that did assess bias (although as far as reviewer #2 is aware, the largest SR assessing bias was my own, and #2 was unable to take bias into account as a subgroup variable due to insufficient reporting).*

This has now been corrected:

“[…]Similarly, previous preclinical systematic reviews have demonstrated that failure to report threats to methodological quality (i.e. internal validity, risk of bias) and construct validity (i.e. extent a model corresponds to the human condition it is intended to represent (Henderson et al., 2013)) influence treatment effect sizes (Crossley et al., 2008; Hirst et al., 2014; Macleod et al., 2015, 2008; Rooke et al., 201119- 23).”

*12) The text states in several places that Mantel Haenszel ORs are used, but the MH method is used to combine several strata specific or study ORs? It is not obvious why an MH OR is needed rather than a crude OR. The meta-analysis is done with a random effects model whereas the MH would be a fixed effects analysis?*

This was the result of a misunderstanding between the first author and our statistical consultant (Ranjeeta Mallick, now listed in acknowledgements). As described in the Methods section and main figures, the meta-analysis was performed with OR and a random effects model. The Supplemental Figure legends (where MH was incorrectly mentioned) have now been corrected.

*13) Pedrazza et al., 2014 – this study shows the same number of deaths in treatment and control- how is it included in the meta-analysis?*

As implied by the reviewers, this data could not be included in the meta-analysis. However, we wanted to display this important data in the total count of animals used in all experiments. We have now clarified this in the Figure legends:

“Data from Pedrazza et al. 2014 was included in total counts but not included in meta-analysis due to 100% mortality in both study arms.”

*14) You are not the first to have used a pre-set list of questions to assess risk of bias and threats to validity and other systematic reviews using similar standards have also poor standards of reporting. Why do you think is it common practice for publications to lack sufficient details of sample size, randomisation and allocation? Is there poor general understanding or appreciation of why it is important to transparently report these details? This could be added to the Discussion.*

These are important questions that we have now addressed in the Discussion:

“[…]A fourth issue is the lack of transparent reporting of risk of bias elements that minimize the ability to evaluate threats to validity in our systematic review. We would suggest that general poor understanding of these core methodological issues may underlie their incomplete reporting. In order to increase the robustness and interpretation of future preclinical systematic review results we submit that authors of primary studies and journal editors should ensure adherence to published reporting guidelines for pre- clinical research studies (National Institutes of Health Principles and guidelines for reporting preclinical research. 2015,Kilkenny, et al., 2010).”

*15) The authors demonstrated that intravenous administration of MSC showed efficacy but intraperitoneal injection did not, and BM or umbilical cord derived- MSC had a better effect than adipose tissue derived MSC. However, only one study (Chang et al. 2012) showed data in an opposite direction in Figure 2—figure supplement 4 and Figure 2—figure supplement 5. Can the authors explain/discuss how the overall efficacy of MSC can be skewed by just a single outlier result?*

This is an excellent point that we have now addressed in the Discussion:

“Overall, subgroup analyses suggested that MSC effects appeared to be robust over a number of varying experimental conditions and across a number of different laboratories. Results of specific subgroups (e.g. autologous cells, multiple doses, intraperitoneal administration, and adipose tissue source) should be interpreted cautiously as few studies were included in these groups, and the results of one study with differing results (Chang et al., 2012) may have skewed data. The ability of one study to heavily influence overall effect estimates is a short-coming of meta-analyses that include few studies.”

*16) At the end of the Introduction it is stated that the study has been written in an explicatory manner so that researchers not familiar with systematic reviews may replicate the approach. The manuscript can be further improved to meet this goal by explaining concepts and including more suggestions for further reading. For example, the authors could more fully explain a funnel plot, used in Figure 4, as this type of analysis may not be known to the audience of "preclinical and translational researchers not familiar with systematic review methodology." Similarly, what is an I2 statistic? What is the role of the PRISMA guidelines?*

These are excellent suggestions that we have now incorporated (largely in the methods section). We have also pointed readers to the methods section to find these details:

Introduction:

“This study has been written in an explicatory manner so that other preclinical and translational researchers not familiar with systematic review methodology may replicate our approach. Readers wishing to replicate our approach for their research agendas are directed to the methods section where explanations are provided in greater depth, and are encouraged to contact the authors for further guidance.”

Methods:

“The methods section has been written completely and transparently for researchers unfamiliar with systematic review and meta-analysis methodology. We would encourage readers wishing to replicate our approach for their own research agendas to refer to available resources (Sena, 2014; Higgins and Green, 2009) and/or contact the authors for further guidance.”

“The protocol for this review was published on the Collaborative Approach to Meta Analysis and Review of Animal Data from Experimental Studies (CAMARADES) website[…]A priori publication of our protocol encourages transparency in the systematic review process and safeguards against reporting biases in the review. This review is reported in accordance with the Preferred Reporting Items for Systematic Reviews and Meta-Analysis (PRISMA) Statement (Spees, Lee and Gregory, 2016). The PRISMA guidelines are an evidence-based minimum set of items that should be reported in a systematic review and meta-analysis. Similar to other reporting guidelines, PRISMA ensures complete and transparent reporting of a study.”

“To identify all relevant studies, we designed a search strategy in collaboration with a medical information specialist. We would suggest to readers consult a medical librarian experienced in systematic searches if they wish to perform a literature search for a preclinical systematic review; this will ensure a comprehensive search is conducted.”

“Statistical analysis was performed in consultation with a statistician experienced in systematic reviews and meta-analysis. Readers seeking to replicate these methods for their own purposes are encouraged to similarly seek advice from an experienced statistician…. Heterogeneity of effect sizes in the overall effect estimates was assessed using the I2 statistic. The following are suggested thresholds to interpret the I2 statistic: 0-40% may not be important, 30-60% moderate heterogeneity, 50-90% substantial heterogeneity, 75-100% considerable heterogeneity (Higgins and Green, 2009).”

“The funnel plot is a scatterplot of the intervention effect of individual studies versus a measure of its precision or size. The characteristic ‘funnel’ shape arises from the fact that precision of the effect estimate increases as the as the study size increases (i.e. small studies will scatter more widely at the bottom of the funnel). A funnel plot would normally be expected to symmetrical, however the absence of symmetry can suggest publication bias (Dominici, 2006). Duval and Tweedie’s trim and fill estimates were generated to estimate the number of missing studies and to estimate the effect estimate assuming the studies were present.”

*17) The authors have clearly shown that there are many significant limitations to analyzing pre-clinical studies due to missing or unclear data. In order to guide future pre-clinical studies, the authors should attempt to provide a "wish list" of variables/data fields they would like to be accounted for in pre-clinical research going forward. This "pre-clinical study blueprint" would be useful in helping to standardize future research efforts so their results can be more easily and accurately aggregated and compared.*

We began to construct a table (appended to this document), however we quickly realized that these concepts would speak to a different audience than we are targeting with the current paper. In the current work we are introducing systematic reviews to researchers who want to translate findings from bench-to- bedside; the table suggested by the reviewers would target basic scientists. The concepts introduced by this table require a great deal more explanation than we can offer in the current paper. We still agree with the underlying sentiment that the reviewers have raised – i.e. more complete and transparent reporting is needed. We have now focused on this in the Discussion:

“In order to increase the robustness and interpretation of future preclinical systematic review results we submit that authors of primary studies and journal editors should ensure adherence to published reporting guidelines for pre-clinical research studies (National Institutes of Health Principles and guidelines for reporting preclinical research. 2015, Kilkenny et al., 2010). These guidelines not only detail items relating to risk of bias (e.g. randomization and blinding) but also touch on issues that are very important when primary studies are included in systematic reviews (e.g. differentiating between biological and technical replicates, providing exact n numbers).”

*18) In the assessment of threats to external validity/generalizability section, two studies were identified that gave multiple injections, one study showing benefit and the other study showing no benefit. Can this discrepancy be clarified by where the MSCs were derived from, as it states earlier that adipose tissue derived MSCs had no significant effect?*

The one study that demonstrated no effect (Chang 2012 et al) was the only one to use autologous cells (harvested from animals 14 days prior to inducing sepsis). This has little construct validity to the clinical setting of sepsis therapy (i.e. pre-identifying patients who will develop severe sepsis is largely impossible, and administration of autologous cells would require 1-2 weeks of expansion initially). There may have been other technical issues that contributed to non-efficacy in this particular study. Clearly, further studies need to be performed to examine the various experimental variables that may contribute to non-efficacy.

We believe the larger issue here is one that is raised by the reviewers above in question #15 – i.e. how one study can skew the results of a subgroup analysis (especially those subgroups that include only two studies). We have addressed this in the Discussion:

“Results of specific subgroups (e.g. autologous cells, multiple doses, intraperitoneal administration, and adipose tissue source) should be interpreted cautiously as few studies were included in these groups, and the results of one study with differing results (Chang et a., 2012) may have skewed data. The ability of one study to heavily influence overall effect estimates is a short-coming of meta-analyses that include few studies.”